# Genetic incorporation of non-canonical amino acid photocrosslinkers in *Neisseria meningitidis*: New method provides insights into the physiological function of the function-unknown NMB1345 protein

Hideyuki Takahashi[1]*, Naoshi Dohmae[2], Kwang Sik Kim[3], Ken Shimuta[1], Makoto Ohnishi[1], Shigeyuki Yokoyama[4,5], Tatsuo Yanagisawa[4,5]

1 National Institute of Infectious Diseases, Department of Bacteriology I, Shinjuku-ku, Japan, 2 Biomolecular Characterization Unit, RIKEN Center for Sustainable Resource Science, Wako, Japan, 3 Division of Pediatric Infectious Diseases, Department of Pediatrics, School of Medicine, Johns Hopkins University, Baltimore, Maryland, United States of America, 4 RIKEN Structural Biology Laboratory, Yokohama, Japan, 5 RIKEN Cluster for Science, Technology and Innovation Hub, Yokohama, Japan

* hideyuki@nih.go.jp

**Data Availability Statement:** All relevant data are within the paper and its Supporting Information

## Abstract

Although whole-genome sequencing has provided novel insights into *Neisseria meningitidis*, many open reading frames have only been annotated as hypothetical proteins with unknown biological functions. Our previous genetic analyses revealed that the hypothetical protein, NMB1345, plays a crucial role in meningococcal infection in human brain microvascular endothelial cells; however, NMB1345 has no homology to any identified protein in databases and its physiological function could not be elucidated using pre-existing methods. Among the many biological technologies to examine transient protein-protein interaction *in vivo*, one of the developed methods is genetic code expansion with non-canonical amino acids (ncAAs) utilizing a pyrrolysyl-tRNA synthetase/tRNA$^{Pyl}$ pair from *Methanosarcina* species: However, this method has never been applied to assign function-unknown proteins in pathogenic bacteria. In the present study, we developed a new method to genetically incorporate ncAAs-encoded photocrosslinking probes into *N. meningitidis* by utilizing a pyrrolysyl-tRNA synthetase/tRNA$^{Pyl}$ pair and elucidated the biological function(s) of the NMB1345 protein. The results revealed that the NMB1345 protein directly interacts with PilE, a major component of meningococcal pili, and further physicochemical and genetic analyses showed that the interaction between the NMB1345 protein and PilE was important for both functional pilus formation and meningococcal infectious ability in *N. meningitidis*. The present study using this new methodology for *N. meningitidis* provides novel insights into meningococcal pathogenesis by assigning the function of a hypothetical protein.

files. The nucleotide sequence of PamA from N. meningitidis strain HT1125 has been deposited in the DNA Data Bank of Japan (DDBJ) (accession code: LC511747).

**Funding:** This work was supported by JSPS KAKENHI Grant Numbers 15K08485 and 19K07550 (to H.T.), AMED under Grant Number 18fk0108071j0101 (to H.T.), by the Platform Project for Supporting in Drug Discovery and Life Science Research (Platform for Drug Discovery, Informatics, and Structural Life Science) from AMED under Grant Number JP16am0101022 (to S.Y.), and under Grant Number JP17am0101081 (to S.Y.).

**Competing interests:** The authors have declared that no competing interest exist.

## Introduction

*Neisseria meningitidis* is a fastidious Gram-negative microorganism that generally exists in the non-invasive so-called "carrier state" at a rate of 0.4–25% in human populations [1, 2]. However, *N. meningitidis* exhibits the abilities to cross the epithelial layer of the upper respiratory tract, infiltrate the bloodstream, evade the defenses of the human immune system, adhere to the endothelial layers of peripheral and brain vessels, cross the brain-blood barrier, and replicate in cerebrospinal fluid [3]. Although epidemiological analyses previously suggested that a human genetic polymorphism [4] and climate conditions [5] are important for predicting the outcomes of infection, the reasons why invasive meningococcal diseases (IMD) occur in some individuals, but not in others remain unclear [6].

Classical analyses revealed the factors exposed on meningococcal surfaces, such as the polysaccharide capsule, pili, Opa, and Opc, (reviewed in [7–9]), and genome mining with whole-genome sequencing (WGS) identified the following meningococcal pathogenic factors: NhhA, NadA [10], App [11], NalP [12], MspA [13], TspA [14], adhesion complex protein (ACP) [15], and fHbp [16]. Among the factors described above, meningococcal pili, which are mainly composed of the major pilin, PilE, and three minor pilins, PilV, PilX, and ComP, play the most important roles in the meningococcal infection processes involved in interactions with human epithelial and endothelial cells (reviewed in [17, 18]). To elucidate the molecular mechanisms underlying meningococcal pathogenesis in more detail, functional genomics by linking genotypes to phenotypes have allowed investigation of the relationships between gene transcript abundance or deficiencies and the capacity to function under various physiological conditions to be investigated, by genome-wide signature-tagged mutagenesis (STM) with an infant rat infection model [19], with human cultured cells [20], and in different media [21], microarrays with infection in human cultured cells [22], comparative genomics [23, 24], transcriptomics [25–28], and proteomics [29–31]. However, the mechanisms by which *N. meningitidis* causes septicemia and meningitis in humans cannot be completely explained by these characterized factors [32].

Although progressive WGS recently identified many genes in the meningococcal genome, more than half of those in the annotated open reading frame have remained as function-unknown hypothetical proteins. One of the difficulties associated with the characterization of meningococcal hypothetical proteins may be the limited number of analytical methods applicable to *N. meningitidis*, including tools to directly manipulate meningococcal components [33, 34]. Under these conditions, we also conducted STM to identify the factors responsible in the sequence type (ST)-2032 *N. meningitidis* strain, known for its high infectious ability in human brain microvascular endothelial cells (HBMEC) [35]. Among the STM mutants with highly defective infectious abilities in HBMEC, we identified PilV [36] of the meningococcal pilus and GltT-GltM glutamate transporter [37, 38] as meningococcal pathogenic factors for infection in host cells. However, several STM mutants were identified as disruptants of hypothetical genes, and we noted that, one disruptant of a hypothetical gene annotated as NMB1345 in the *N. meningitidis* MC58 genome [39], exhibited largely defective internalization ability into HBMEC. However, the deduced amino acid sequence of NMB1345 lacked homology with any function-known proteins from other species. We previously attempted to examine protein-protein interactions using already existing methods, such as the two-hybrid system or a pull-down assay, to find some clues about the function of the NMB1345 protein but were unsuccessful. Therefore, more powerful strategies are needed to elucidate the function of the NMB1345 protein in *N. meningitidis*.

One of the appropriate methods for our study is genetic code expansion with non-canonical amino acids (ncAAs) utilizing a pyrrolysyl-tRNA synthetase/tRNA$^{Pyl}$ pair from *Methanosarcina mazei* (MmPylRS/T) [40]. The genetic code expansion allows ncAAs containing fluorophores, photocaged groups, photocrosslinkers, metal ion-chelating groups, and other groups

to be site-specifically incorporated into target proteins *in vivo* in a broad range of species from *Escherichia coli* to eukaryotic systems, such as *Caenorhabditis elegan*s, fruit flies, and mice (reviewed in [41]), thereby enabling *in vivo* examinations of biological functions. Protein photocrosslinking via the incorporation of photoreactive ncAAs has emerged as a superior strategy for identifying physiological interaction partner(s) and their functions [42, 43], and in combination with mass spectrometry (MS), these approaches have resolved biological issues that are difficult or impossible to address using the majority of currently available methods [42]. However, the question of whether the orthogonality of the tRNA$^{Pyl}$-PylRS pair is specific or compatible with the translational system in *N. meningitidis* has not been addressed.

In the present study, we applied genetic code expansion to *N. meningitidis* and developed a method for the genetic incorporation of photoreactive ncAAs into *N. meningitidis* in order to identify the physiological function of the NMB1345 protein. After establishing this method, we performed site-specific incorporation with one of the most widely used photoreactive amino acids, *p*-benzoyl-L-phenylalanine (*p*BPa) [44], into the NMB1345 protein in *N. meningitidis*. By using crosslinking in *N. meningitidis* followed by purification and MS analyses of endogenous proteins crosslinked to the NMB1345 protein, we found that the NMB1345 protein specifically interacted with PilE, a major component of meningococcal pili, and thus designated NMB1345 as PilE associated molecule A (PamA) in the present study. We further identified the amino acid residue of PilE crosslinked to PamA K273 and further found that the site-directed PilE mutation reduced both the interaction with PamA and the ability to internalize into HBMEC. Considering these results, we speculated that PamA plays an important role in the formation of meningococcal pili. This is the first study to successfully incorporate ncAAs into *N. meningitidis* in order to determine the physiological function of a protein with an unknown function and structure in pathogenic bacteria.

## Materials and methods

### Bacterial growth conditions

*N. meningitidis* strains HT1125 [35] and its derivatives, H44/76 [45] as well as derivatives harboring IncQ plasmids (Table 1) were routinely grown on GC agar plates at 37 $^{o}$C in a 5% $CO_2$ atmosphere [46]. *E. coli* strains were grown in Luria-Bertani (LB) broth liquid cultures (Becton- Dickinson) or on LB plate (LB liquid medium containing 1.5% agar) or in MagicMedia (Invitrogen) at the indicated temperatures. When required, antibiotics were added at the following concentrations: kanamycin at 150 μg/ml, chloramphenicol at 5 μg/ml, erythromycin at 4 μg/ml, and spectinomycin at 75 μg/ml for *N. meningitidis*; kanamycin at 50 μg/ml, chloramphenicol at 25 μg/ml, and ampicillin at 50 μg/ml for *E. coli*. The *N. meningitidis* and *E. coli* strains used in the present study are listed in Table 2.

### Production of anti-PamA protein rabbit antiserum

All PCR experiments were performed with PrimeSTAR Max GXL DNA polymerase (Takara Bio, Japan). The approximately 3.5-kb PCR fragment containing the *pamA* gene amplified from *N. meningitidis* HT1125 genomic DNA with the primer set (NMB1345-21 and NMB1345-2) (S1 Table) was cloned into the pTWV228 vector (Takara Bio, Japan), resulting in pHT922 (S2 Table), which is the ancestral plasmid in the present study. A *ΔN-pamA* gene, in which the N-terminal hydrophobic region consisting of the first 23 amino acids was removed (S1 Fig), was amplified by PCR with the primer set (NMB1345(pET303)-1(NsiI) and NMB1345(pET303)-2(BH)) (S1 Table) and cloned into the same site of the pET303/CT-His expression plasmid (Invitrogen) to construct pHT934 (S3 Table). Regarding the production of the ΔN-PamA protein for rabbit immunization, a 1 L culture of *E. coli* strain BL21 (NEB)

**Table 1. IncQ plasmids introduced into *N. meningitidis* strains which results were shown in this study.**

| Plasmid | Relative properties | Antibiotic selection marker | References |
|---|---|---|---|
| pHT128 | Derivative of pGSS33, an IncQ broad-host -range vector | Tet, Cml | [33] |
| pHT1212 | Derivative of pHT128 carrying a $lacI^q$-$P_{tac}$-MmPylRS[Y306A/Y384F] and two $P_{lpp}$-tRNA$^{pyl}$ genes | Tet, Cml | This study |
| pHT1262 | Derivative of pHT128 carrying a $lacI^q$-MmPylRS([A302T/N346T/C348T/W417C]) and two $P_{lpp}$-tRNA$^{pyl}$ genes | Tet, Cml | This study |
| pHT936 | Derivative of pHT128 carrying $lacI^q$-$P_{tac}$-$gst^+$ genes | Cml | This study |
| pHT1355 | Derivative of pHT1212 carrying a $P_{tac}$-gst E51amb gene | Cml | This study |
| pHT1356 | Derivative of pHT1212 carrying a $P_{tac}$-gst F52amb gene | Cml | This study |
| pHT1357 | Derivative of pHT1262 carrying a $P_{tac}$-gst E51amb gene | Cml | This study |
| pHT1358 | Derivative of pHT1262 carrying a $P_{tac}$-gst F52amb gene | Cml | This study |
| pHT1263 | Derivative of pHT1262 carrying a $pamA^+$-$His_6$ gene | Cml | This study |
| pHT1270 | Derivative of pHT1262 carrying a pamA K208amb-$His_6$ gene | Cml | This study |
| pHT1274 | Derivative of pHT1262 carrying a pamA K278amb-$His_6$ gene | Cml | This study |
| pHT1276 | Derivative of pHT1262 carrying a pamA K309amb-$His_6$ gene | Cml | This study |
| pHT1279 | Derivative of pHT1262 carrying a pamA K341amb-$His_6$ gene | Cml | This study |
| pHT1282 | Derivative of pHT1262 carrying a pamA K382amb-$His_6$ gene | Cml | This study |
| pHT1284 | Derivative of pHT1262 carrying a pamA K395amb-$His_6$ gene | Cml | This study |
| pHT1388 | Derivative of pHT1262 carrying a pamA K278amb-StrepTag$_2$-$His_6$ gene | Cml | This study |

Tet and Cml stand for tetracycline and chloramphenicol resistance marker, respectively.

harboring the plasmid was grown in MagicMedia at 37°C overnight, and the cells were collected. The subsequent purification of the recombinant protein and generation of polyclonal rabbit antibodies to the putative hydrophilic domain of the PamA protein were performed as described previously [46].

## Construction of meningococcal mutants

In order to construct the *N. meningitidis pamA* deletion mutant, the 6.3-kb PCR fragment was amplified with the primers (NMB1345-3 and NMB1345-4) (S1 Table) from pHT922, in which a 1.5-kb DNA fragment containing the coding region of the *pamA* gene was completely removed, and ligated with a spectinomycin-resistance gene (*spc*) after phosphorylation to construct pHT930. A 3-kb DNA fragment, in which the *pamA* structural gene was replaced with the *spc* gene, was amplified with the primers (NMB1345-21 and NMB1345-2) (S1 Table) from pHT930 and transformed into *N. meningitidis* strain HT1125, and spectinomycin-resistant (Spc$^R$) clones were selected, resulting in the *ΔpamA* mutant HT1822 (Table 2).

The *ΔpamA* mutant complemented with the *pamA$^+$* gene at the *ggt* allele was constructed as follows: A chloramphenicol-resistance gene (*cat*) was inserted into the SmaI site downstream of the *pamA* gene of pHT922 to generate pHT923 (S2 Table). A 4.6-kb PCR fragment containing the *pamA$^+$-cat* gene was amplified from pHT923, in which the meningococcal *ggt* structural gene was cloned in pTWV228 with primers (M13-RV-ggt-5′and M13-47-ggt-3′) (S1 Table) and cloned into the BstXI site of pHT195 (at the middle of the *ggt* coding region) by In-Fusion cloning (Clontech), resulting in the plasmid pHT924. A 6.3-kb DNA fragment containing the *ggt::pamA$^+$-cat* genes was amplified from pHT924 with primers (ggt-3 and ggt-4) (S1 Table), transformed into *N. meningitidis ggt::pamA$^+$-cat* mutants, and chloramphenicol-resistant (Cml$^r$) clones were selected, resulting in the *pamA* deletion mutant HT2224, which were ectopically complemented with the *pamA$^+$* genes expressed from their own promoter at the *ggt* allele (Table 1).

Since we have demonstrated that the *pilE-cat* translational fusion could work as well as the wild-type *PilE$^+$ N. meningitidis* strain [36], an *N. meningitidis* I12A *pilE* mutant, in which Ile at

**Table 2. Strains used in this study.**

*N. meningitidis* strains

**HT1125**

| Strain | Genotype | Parent strain | Reference |
|---|---|---|---|
| HT1125 | *ΔsiaB-ΔsiaD::kan* (ST-2032) | [35] | |
| HT1572 | *ΔsiaB-ΔsiaD::kan pamA::Tn-spc* | HT1125 | This study |
| HT1736 | *ΔsiaB-ΔsiaD::kan pamA::Tn-spc ggt::pamA$^+$-cat* | HT1572 | This study |
| HT1822 | *ΔsiaB-ΔsiaD::kan ΔpamA::spc* | HT1125 | This study |
| HT2224 | *ΔsiaB-ΔsiaD::kan ΔpamA::spc ggt::pamA$^+$-cat* | HT1822 | This study |
| HT2215 | *ΔsiaB-ΔsiaD::kan pamA'-lacZ-ermC* | HT1822 | This study |
| HT2217 | *ΔsiaB-ΔsiaD::kan pamA'-phoA-ermC* | HT1822 | This study |
| HT1744 | *ΔsiaB-ΔsiaD::kan pilE$^+$-cat* (translational fusion) | HT1125 | This study |
| HT2167 | *ΔsiaB-ΔsiaD::kan pilE I12A-cat* (translational fusion) | HT1125 | This study |
| HT2218 | *ΔsiaB-ΔsiaD::kan pilF$^+$-FLAG-ermC* | HT1125 | This study |
| HT2219 | *ΔsiaB-ΔsiaD::kan pilF$^+$-FLAG -ermCΔpamA::spc* | HT1822 | This study |
| HT2136 | *ΔsiaB-ΔsiaD::kan pilP$^+$-FLAG -ermC* | HT1125 | This study |
| HT2137 | *ΔsiaB-ΔsiaD::kan pilP$^+$-FALG-ermCΔpamA::spc* | HT1822 | This study |
| HT2132 | *ΔsiaB-ΔsiaD::kan pilM$^+$-HA -ermC* | HT1125 | This study |
| HT2133 | *ΔsiaB-ΔsiaD::kan pilM$^+$-HA -ermC ΔpamA::spc* | HT1822 | This study |
| HT2211 | *ΔsiaB-ΔsiaD::kan pilX$^+$-FLAG-ermC* | HT1125 | This study |
| HT2212 | *ΔsiaB-ΔsiaD::kan pilX$^+$-FLAG -ermC ΔpamA::spc* | HT1822 | This study |
| HT1688 | *ΔsiaB-ΔsiaD::kan pilE::ermC* | HT1125 | [36] |
| HT1822 | *ΔsiaB-ΔsiaD::kan ΔpilV::ermC* | HT1125 | [36] |
| HT2230 | *ΔsiaB-ΔsiaD::kan pilE$^+$-cat pilX$^+$-FLAG-ermC* | HT1744 | This study |
| HT2231 | *ΔsiaB-ΔsiaD::kan pilE I12A-cat pilX$^+$-FLAG-ermC* | HT2167 | This study |

**H44/76**

| Strain | Genotype | Parent strain | Reference |
|---|---|---|---|
| H44/76 | wild type strain (ST-32, Serogroup B) | | [33] |
| HT1001 | H44/76 *gyrA* (Nal$^R$) | H44/76 | This study |
| HT1940 | *ΔpamA::spc* | H44/76 | This study |
| HT2095 | *ΔpamA::spc pilE$^+$-FLAG-ermC* | HT1940 | This stud |
| HT2014 | *ΔpamA::spc pilE::ermC* | HT1940 | This study |
| HT2162 | *ΔpamA::spc pilE$^+$-ermC* (translational fusion) | HT1940 | This study |
| HT2166 | *ΔpamA::spc pilE I12A-ermC* (translational fusion) | HT1940 | This study |
| HT1015 | *pilE::ermC* | H44/76 | [35] |

*Escherichia coli* strains

| Strain | Genotype | | References |
|---|---|---|---|
| JM109 | *endA1 gyrA96 hsdR17(rk⁻mk$^+$) mcrB$^+$ recA1 relA1 supE44 thi-1Δ(lac-proAB) F'[traD36 proAB lacI$^q$ZΔM15]* | | Nippon gene |
| BL21 | F−*dcm ompT hsdSB(rB−mB−) gal* | | NEB |
| BL21(DE3) | F−*dcm ompT hsdSB(rB−mB−) gal λ(DE3)* | | NEB |

position 12 was substituted with Ala, was constructed as described previously. In brief, pHT872 (*pilE$^+$-cat* translational fusion on pGEM-3z) (S2 Table) [36] was site-directed muta-genized with primers (pilE-I12A-1 and pilE I12A-2) (S1 Table) to construct pHT1536 (S2 Table). After confirming its sequence, a 1.5-kb PCR fragment containing the *pilE I12A-cat* gene was amplified from pHT1536 by a primer set (T7 and SP6) (S1 Table) and transformed into HT1125. Cml$^r$ clones were selected and the mutation was confirmed by direct sequencing as for the *pilE I12A-cat N. meningitidis* mutant HT2167 (Table 1).

To construct the *pilE⁺-ermC* or *pilE I12A-ermC* translational fusion in *N. meningitidis* strain H44/76 harboring the IncQ plasmid, a 0.9-kb DNA fragment from HT1125 chromosomal DNA containing the structural gene of the wild-type *pilE* gene and its upstream region was amplified with both primers (pGEM-3z-(SmaI)-2(15mer)-pilE-51 and pilE-52) (S1 Table). A 0.2-kb DNA fragment of the *pilE* downstream region was also amplified with a primer set (pilE-53 and pilE-18) (S1 Table), and a 0.75-kb DNA fragment of the *ermC* gene containing the SD sequence was amplified from pHT24 with a primer set (ermC-21 and ermC-22) (S1 Table). The three PCR fragments were cloned into the SmaI site of pGEM-3z to construct pHT1504 (S2 Table). The *N. meningitidis* I12A *pilE* mutant was constructed as described above using the primer set (pilE-I12A-1 and pilE I12A-2) (S1 Table) with pHT1504 as the template. After the sequence was confirmed, a 1.5-kb PCR fragment containing the *pilE I12A-ermC* genes was amplified from pHT1536 (S2 Table) with universal primers (T7 and SP6) and transformed into HT1125. Erm<sup>r</sup> clones were selected and the mutation was confirmed by direct sequencing as for the *pilE I12A-ermC N. meningitidis* mutant HT2166 (Table 1). A 1.5-kb PCR fragment containing the *pilE⁺-ermC* genes amplified from pHT1504 (S2 Table) or the *pilE I12A-ermC* genes amplified from pHT1536 (S2 Table) by universal primers (T7 and SP6) was transformed into HT1940. Erm<sup>r</sup> clones were selected and the mutation was confirmed by direct sequencing as for the *pilE⁺-ermC* HT2162 or *pilE I12A-ermC* HT2166 *N. meningitidis* mutants in the *ΔpamA::spc* genetic background (Table 1).

The addition of a FLAG tag to the *pilE* gene at the 3′-terminus on its chromosomal locus was achieved as follows: a 1.6-kb PCR fragment containing the *pilE* allele in H44/76 was amplified with a primer set (pilE-11and pilE-12) (S1 Table), and after phosphorylation, it was cloned into pMW119 to construct pHT1419 (S2 Table). A 5.8-kb PCR fragment was amplified with a primer set (pilE-17 and FLAG'(15mer)-pilE-18) (S1 Table) from pHT24, and a 1-kb PCR fragment containing the *ermC* gene was also amplified with a primer set (pilE-17'(15mer)-M13-47 and FLAG-RV-M) (S1 Table) from pHT24. The two PCR fragments were ligated by In-Fusion Cloning to construct pHT1420, in which the FLAG tag was fused to the *pilE* structural gene followed by an *ermC* selection marker (S2 Table). A 2.6-kb PCR fragment containing the *pilE⁺-FLAG-ermC* genes was amplified from pHT1420 with a primer set (pilE-11 and pilE-12) and transformed into HT1940. Erm<sup>r</sup> clones were selected and the mutation was confirmed by direct sequencing as for the *pilE⁺-FLAG-ermC N. meningitidis* mutant HT2095 (Table 1).

The *pilE* insertional mutant HT2014 was constructed by transforming HT1940 with 500 ng of HT1015 (H44/76 *pilE::ermC*) [35] chromosomal DNA, followed by the selection of the Erm<sup>r</sup> strain HT2014 (Table 1).

The addition of a FLAG tag to the *pilF* gene at the 3′-terminus on its chromosomal locus was performed as follows: a 1.8-kb PCR fragment containing the *pilF* allele in the H44/76 *N. meningitidis* strain was amplified with a primer set (pMW(SmaI)-up(15mer)-pilF-1 and pMW(SmaI)-down(15mer)-pilF-2) (S1 Table) and cloned into the SmaI site of pMW119 by In-Fusion cloning to construct pHT1449 (S1 Table). A 6-kb PCR fragment amplified with a primer set (pilF-3 and FLAG'(15mer)-pilF-4) (S1 Table) from pHT1499 and a 1-kb PCR fragment containing the *ermC* gene amplified with a primer set (pilF-3'(15mer)-M13-47 and FLAG-RV-M) (S1 Table) from pHT24 were ligated by In-Fusion Cloning to construct pHT1457, in which the FLAG tag was fused to the *pilF* structural gene followed by an *ermC* selection marker (S1 Table). A 2.8-kb PCR fragment containing the *pilF⁺-FLAG-ermC* genes was amplified from pHT1457 with a primer set (pMW(SmaI)-up(15mer)-pilF-1 and pMW(SmaI)-down(15mer)-pilF-2) (S1 Table), and then transformed into HT1125 and HT1822. Erm<sup>r</sup> clones were selected and the mutation was confirmed by direct sequencing as for the *pilF⁺-FLAG-ermC N. meningitidis* mutants HT2218 and HT2219, respectively (Table 1).

A 3.9-kb DNA fragment containing partial *pilM-pilN⁺-pilO⁺-pilP⁺* genes in H44/76 *N. meningitidis* was amplified with a primer set (pMW(SmaI)-up(15mer)-pilMNOP-3 and pMW(SmaI)-down(15mer)-pilMNOP-4) (S1 Table) and cloned into the SmaI site of pMW119 by In-Fusion cloning to construct pHT1450 (S2 Table).

The addition of a FLAG tag to the *pilP* gene at the 3' terminus on its chromosomal locus was achieved as follows: a 7.2-kb DNA fragment amplified with a primer set (pilP-3 and FLAG'(15mer)-pilP-2) (S1 Table) from pHT1450 and a 1-kb PCR fragment containing the *ermC* gene amplified with a primer set (pilP-3'(15mer)-M13-47 and FLAG-RV-M) (S1 Table) from pHT24 were ligated by In-Fusion cloning to construct pHT1454 (S2 Table), in which the FLAG tag was fused to the *pilP* structural gene at the 3' terminus followed by an *ermC* selection marker. A 4.9-kb PCR fragment containing the *pilP⁺-FLAG-ermC* genes was amplified from pHT1454 with a primer set (pMW(SmaI)-up(15mer)-pilMNOP-3 and pMW(SmaI)-down(15mer)-pilMNOP-4)(S1 Table) and transformed into HT1125 and HT1822. Erm^r clones were selected and the mutation was confirmed by direct sequencing as for the *pilP⁺-FLAG-ermC N. meningitidis* mutants HT2136 and HT2137, respectively (Table 1).

The addition of the HA tag to the *pilM* gene at the 3' terminus on its chromosomal locus was achieved; however, the ORFs of the *pilM* (1.1 kb), *pilN* (0.6 kb), and *pilO* (0.6 kb) genes were very close or overlapped, and thus an *ermC* selection marker was inserted downstream of the *pilO* gene. A 7.1-kb PCR fragment containing the partial *pilM* gene and pMW119 from pHT1450 was amplified with a primer set (pilO-3 and HA'-pilM-2) (S1 Table). A 1.2-kb PCR fragment was amplified with a primer set (HA'(15mer)-pilN-1 and pilO-2) from pHT1450 and a 1-kb *ermC* gene fragment was also amplified with a primer set (pilO-2'(15mer)-M13-RV and pilO-3(15mer)-M13-47) (S1 Table) from pHT24. The three PCR fragments were ligated by In-Fusion Cloning to construct pHT1453, in which the HA tag was fused to the *pilM* structural gene with an *ermC* selection marker (S2 Table). A 2.6-kb PCR fragment containing the *pilM⁺-HA pilN⁺ pilO⁺-ermC-pilP* genes was amplified from pHT1453 with a primer set (pMW(SmaI)-up(15mer)-pilMNOP-3 and pMW(SmaI)-down(15mer)-pilMNOP-4) (S1 Table) and transformed into HT1125 and HT1822. Erm^r clones were selected and the mutation was confirmed by direct sequencing as for the *pilM⁺-HA-ermC N. meningitidis* mutants HT2132 and HT2133, respectively.

The addition of a FLAG tag to the *pilX* gene at the 3' terminus on its chromosomal locus was achieved as follows: a 0.5-kb *pilX* DNA region containing the 400-bp upstream and 160-bp downstream regions of the *pilX* ochre codon in H44/76 was amplified with a primer set (pMW(SmaI)-up(15mer)-pilX-3 and pMW(SmaI)-down(15mer)-pilX-6) (S1 Table), and then cloned into the SmaI site of pMW119 by In-Fusion cloning to construct pHT1594. A 4.7-kb PCR fragment amplified with a primer set (M13-47'(15mer)-pilX-5 and FLAG'(15mer)-pilX-8) (S1 Table) from pHT1594, and a 1-kb PCR fragment containing the *ermC* gene amplified with a primer set (FLAG-RV-M and M13-47) (S1 Table) from pHT24 were ligated by In-Fusion Cloning to construct pHT1597, in which the FLAG tag was fused to the *pilX* structural gene followed by an *ermC* selection marker (S2 Table). A 1.5-kb PCR fragment containing partial *pilX⁻ FLAG-ermC* genes was amplified from pHT1597 with a primer set (pMW(SmaI)-up(15mer)-pilX-3 and pMW(SmaI)-down(15mer)-pilX-6) (S1 Table) and transformed into HT1125 and HT1822. Erm^r clones were selected and the mutation was confirmed by direct sequencing as for the *pilX⁺-FLAG-ermC N. meningitidis* mutants HT2211 and HT2212, respectively (Table 1).

*pamA-lacZ* translational fusion *N. meningitidis* mutants were constructed as follows: a 3-kb PCR fragment containing the *lacZ* gene was amplified from *E. coli* strain W3110 [47] chromosomal DNA with the primers (lacZ-21 and M13-RV'(15mer)-lacZ-22) (S1 Table) and a 1-kb *ermC* gene fragment was amplified with universal primers (M13-RV and M13-47) (S1 Table)

from pHT24 [33]. The two PCR fragments were cloned into a 7.2-kb DNA fragment (designated Fragment C) amplified from pHT922 containing the upstream and downstream regions of the *pamA* gene with the first 408-bp region including the N-terminal membrane spanning region (designated *pamA′*) with a primer set (M13-47′(15mer)-nmb1345-61 and lacZ′-21 (15mer)-nmb1345-62) (S1 Table) by In-Fusion cloning to construct pH1605 (S2 Table and Fig 2B).

*pamA′-phoA* translational fusion *N. meningitidis* mutants were constructed as follows: a 1.5-kb PCR fragment containing the *phoA* gene was amplified from *E. coli* strain W3110 chromosomal DNA with the primers (phoA-21 and M13-RV'(15mer)-phoA-2) (S1 Table), and a 1-kb *ermC* gene fragment was amplified with universal primers (M13-RV and M13-47) from pHT24 [33]. The two PCR fragments were cloned into Fragment C by In-Fusion Cloning to construct pH1603 (S2 Table). PCR fragments amplified with nmb1345-5 and nmb1345-21 (S1 Table) from pHT1605 containing the *pamA′-lacZ-ermC*-downstream region of the *pamA* gene or from pHT1603 containing the *pamA′-phoA-ermC*-downstream region of the *pamA* gene were transformed to HT1125, and erythromycin-resistant clones were selected, as for the *pamA′-lacZ*::*ermC* or *pamA′-phoA*::*ermC N. meningitidis* mutants on its chromosome HT2215 and HT2217, respectively (Table 1 and Fig 2B).

## Assessment of host cell-associated and internalized bacteria

The infection of HBMEC with *N. meningitidis* was performed as described previously [46]. Results are expressed as means ± the standard deviation (SD), and bacterial numbers were statistically compared using the two-tailed Student's *t*-test.

## Western blotting

SDS-PAGE and Western blotting were performed as described previously [48].

## Observations of meningococci and ezrin accumulation by immunofluorescence staining

HBMEC monolayer preparations, *N. meningitidis* infections of these layers, and the preparation of immunostaining samples were performed as described previously [46]. Infected HBMEC and attached meningococci were observed using an ECLIPSE E600 microscope (Nikon) with a 100× oil immersion objective.

## Construction of IncQ plasmids for *N. meningitidis*

The mutated *M. mazei pylT* (tRNA^Pyl) and mutated *pylRS* genes for *m*AzZLys (*PylRS*[Y306A/Y384F]) [49] and *p*BPa (*PylRS* A302T/N346T/C348T/W417C]) [50] (*MmpBpaRS*) were derived from pCDF-Pyl-Fx1 [49]. The four point mutations were generated by three-step mutations with the following three primer sets (A302-T and A302-7-R) (S1 Table), (N346T-C348T-F and N346-C348-R) (S1 Table), and (W417C-F and W417C-R) (S1 Table) using the Mutagenesis Basal Kit (Takara Bio). The plasmid constructed from pCDF-Pyl-Fx1 as described above was named pCDF-Pyl-Fx3 in this study (S3 Table). A 2-kb PCR fragment containing one copy of the *pylRS* gene (Y306A/Y384F) and two copies of the *pylT* genes expressed from the *E. coli lpp* promoter [49] (*pylRS (Y306A/Y384F)/T*) from pCDF-Pyl-Fx1 or a 2-kb PCR fragment containing one copy of the *MmpBPaRS* genes and two copies of the *pylT* genes expressed from the *E. coli lpp* promoter [49] (*MmpBPaRS/T*) from pCDF-Pyl-Fx3 were amplified by PCR with a primer set (ptac-12-pylRS-1 and M13-47-reverse-lppT-2) (S1 Table), and then cloned into pTTQ18, an expression vector with a strong *tac* promoter ($P_{tac}$) controlled by

the LacI protein [51] using In-Fusion cloning to construct pHT1208 and pHT1261, respectively (S2 Table). The PCR fragment containing the *lacI$^q$-P$_{tac}$-pylRS[Y306A/Y384F]/T* or *lacI$^q$-P$_{tac}$-MmpBPaRS/T* genes was amplified by PCR with primers (M13-47 reverse and ptac-12) (S1 Table), and after phosphorylation, the fragment was cloned into the blunted *Sac*I and *Pst*I sites of pHT128, an IncQ plasmid [33], resulting in pHT1212 (*lacI$^q$-P$_{tac}$-pylRS[Y306A/Y384F]/ T*) and pHT1262 (*lacI$^q$-P$_{tac}$-MmpBPaRS/T*), respectively (Table 1 and S4 Table).

PCR fragments containing the *gst* gene derived from pGEX-6P1 (GE Healthcare) with an amber codon at position Glu 51 (*gst E51*) or Phe 52 (*gst F52*) were constructed as follows: the *gst E51* or *gst F52* mutation was introduced by site-directed mutagenesis with two sets of primers (gst E51amb-1 and gst E51amb-2 for *gst E51)* (gst F52amb-1 and gst F52amb-2 for *gst F52)* (S1 Table) using the PrimeSTAR Mutagenesis Basal Kit. After the sequence was confirmed, a 0.5-kb PCR fragment containing *Ptac-gstE51amb* or *Ptac-gstF52amb* was amplified with a primer set (tetA-1-gst-up and gst-down-tetA-2) (S1 Table) and cloned into the SalI site of pHT1212 or pHT1216 to construct pHT1355, pHT1356, pHT1357, and pHT1358, respectively (Table 1).

To construct a series of amber mutations in the *pamA* gene, the His$_6$ tag was initially introduced into the 3' terminus of the native *pamA* gene as follows: a 7.2-kb PCR fragment was amplified from pHT922 with a primer set (NMB1345-1 and NMB1345-2-(His)$_6$) (S1 Table), phosphorylated, and then self-ligated to construct pHT1213 (Table 1).

Amber mutation series (*pamA K-amb*) plasmids (see S1 Fig) were constructed by site-directed mutagenesis of pHT1213 (S2 Table) with primer sets (S1 Table). The *pamA K-amb* gene on pTWV228 was confirmed by sequencing and the plasmid is listed in S2 Table. A 2.2-kb PCR fragment containing the *pamA K-amb* gene was amplified with a primer set (tetA-NMB1345-up and tetA-NMB1345-down) (S1 Table) from the pTWV228-based plasmids listed in S2 Table, and cloned into the SalI site of pHT1216 or pHT1262 by In-Fusion cloning to construct IncQ plasmids containing the *lacI$^q$-P$_{tac}$-pylRS[Y306A/Y384F]/T* and *pamA K-amb* genes (pHT1212) or *lacI$^q$-P$_{tac}$-MmpBPaRS/T* and *pamA K-amb* genes (pHT1262) listed in Table 1.

The addition of *StrepTag$_2$-His$_6$* synthetic DNA (S3 Fig) to the *pamA K278amb* gene was achieved as follows: a 6.5-kb PCR fragment amplified from pHT1242 with a primer set (spacer-1(15mer)-nmb1345-12 and His$_6$'(15mer)-nmb1345-11) (S1 Table) was ligated to *StrepTag$_2$-His$_6$* synthetic DNA by In-Fusion cloning to construct pHT1293 (S2 Table). A 2.4-kb PCR fragment containing the *pamAK278amb-StrepTag$_2$-His$_6$* gene was amplified with a primer set (tetA-NMB1345-up and tetA-NMB1345-down) (S1 Table) from pHT1293 and cloned into the SalI site of pHT1262 by In-Fusion cloning, to construct pHT1388 (Table 1).

The introduction of IncQ plasmids into *N. meningitidis* strains was performed as described previously [33].

## UV crosslinking of whole *N. meningitidis* cells expressing the PamA Lys-amber (*pamA K-amb*) mutants by pyrrolysine-based amber suppression with *p*BPa

The *N. meningitidis* H44/76 *ΔpamA::spc* mutant (HT1940) (Table 2) harboring IncQ plasmids containing *lacI$^q$ -Ptac-MmpBPaRS/T* genes and *pamA K-amber* genes (Table 1) was cultivated on GC agar plates containing 1 mM IPTG and 1 mM *p*BPa at 37˚C in 5% CO$_2$ for 18 hours. To identify the optimal *pamA K-amb* mutants for pyrrolyl-based amber suppression followed by photocrosslinking, *N. meningitidis* transformants grown one-fourth of a GC plate were suspended in 50 μl PBS, transferred into the wells of a 96-well plate/strain, and irradiated with 365 nm UV using a FL-365-SD UV lamp stand (Opticode, Japan) at a distance of 2 cm on ice

for 2 hours. After harvesting, UV-irradiated bacteria were resuspended in 400 μl of urea buffer (50 mM Tris-HCl pH7.5, 8 M urea, 1% Triton X-100, and 100 mM NaCl) and then transferred into a 1.5-mL tube. Bacteria were disrupted by sonication, and the soluble fraction was obtained by centrifugation. PamA K-amb-His$_6$ proteins with expression suppressed by *p*BPa were purified with 200 μL TALON resin (Clontech). The purified fraction (approx. 250 μl) was precipitated by TCA/acetone at -80 $^\circ$C overnight, and the precipitate was finally dried and suspended in 25 μl 1 × SDS buffer. Aliquots were analyzed by SDS-PAGE.

## Identification of endogenous proteins crosslinked with PamA K278(*p*BPa) in *N. meningitidis*

To identify endogenous protein(s) crosslinked to PamA K278(*p*BPa) in *N. meningitidis*, an synthetic DNA fragment containing twin *Strep*-Tag (StrepTag$_2$) [52] was further introduced into pHT1274, resulting in pHT1388 (S3 Fig and Table 2). *N. meningitidis* HT1940 harboring pHT1388 was grown on approximately 200 GC agar plates containing 1 mM IPTG and 1 mM *p*BPa at 37˚C in 5% CO$_2$ for 18 hours. The bacteria from 3 GC agar plates were suspended in 0.5 mL PBS, transferred into the wells of a 12-well plate (more than 50 wells in total), and irradiated with 365 nm UV on ice for 2 hours. The irradiated bacteria from 2 wells were transferred into a 15-mL tube and harvested by centrifugation. The collected bacteria were suspended in 5 mL IP Lysis buffer (25 mM Tris-HCl pH7.5, 150 mM NaCl, 1 mM EDTA, 1% NP-40, and 5% glycerol) and disrupted by sonication followed by centrifugation at 12,000 rpm at 4˚C for 30 min. The supernatant (soluble fraction) was collected into one tube. Crosslinked proteins were purified by two column chromatography steps using Ni-Sepharose High Performance (GE Healthcare) and *Strep*-Tactin Sepharose (Iba, Germany). The purified fraction was concentrated approximately 70-fold using a VIVASPIN TURBO 15 (MW 50k) Centrifuge Filter unit (Sartorius), to a final volume of approximately 100 μL. An aliquot of the sample was analyzed by 4–12% NuPAGE (Invitrogen), and immunoblotting with an anti-His$_6$ monoclonal antibody (mAb) (Wako, Japan). All proteins were also visualized using the CBB staining kit (APRO SCIENCE, Japan) and bands that were also detected by immunoblotting were excised. These bands were subjected to trypsin digestion and peptide extraction. Extracted peptides were dried and resuspended in H$_2$O and 0.1% (w/w) trifluoroacetic acid for MS. The LC-MS/MS analysis was performed with an Advance nanoLC (Michrom Biosources) coupled to an LTS Mass Spectrometer (Thermo Fisher Scientific). Proteins were identified by running MAS-COT (http://www.matrixscience.com) against the NCBI database (NCBInr 20160202). Candidates with molecular masses that were consistent with the difference between the crosslinked complexes and the PamA protein were selected. The accuracy of each interaction between PamA K278(*p*BPa) and the candidate was further examined by constructing *N. meningitidis* HT1940 harboring pHT1388 mutants, in which the FLAG tag was added to the candidate gene on its chromosome, as in the *pilX-FLAG N. meningitidis* strain (Table 1) described above.

## Identification of the counterpart of the amino acid residue of PilE crosslinked with PamA K278(*p*BPa)

A DNA fragment containing *ΔN-pamA K278amb* was amplified from pHT1242 with a primer set (NMB1345(pGEX)-1(BH) and NMB1345(pGEX)-2(XhoI)) (S1 Table) and cloned into the same sites of pMAL-c2 (NEB) to construct pHT1473 (S3 Table). A 0.5-kb PCR fragment containing the Gln$_6$-ΔN-PilE gene, in which the first 9 amino acids (PilE leader peptides in *N. meningitidis*) were replaced with 6 glutamine residues (Gln$_6$) (see the Results section), was amplified with a primer set ((Gln)$_6$-pilE-45 and pCDF-1b(PacI)(15mer)-pilE-44) (S1 Table) from H44/76 *N. meningitidis* chromosomal DNA, and designated as fragment B. To clone the

PCR fragment into pCDF-1b-kan (S3 Table) [53], a 3.6-kb DNA fragment was amplified from pCDF-1b-kan with a primer set (pCDF-1b (downstream of PacI) and $(Gln)_6$(15mer)-pCDF-1b-2) (S1 Table) and ligated to PCR fragment B by In-Fusion Cloning to construct pHT1447 (S3 Table), in which the $\Delta N$-$(Gln)_6$-PilE protein was expressed from the T7 promoter ($P_{T7}$). A 0.6-kb PCR fragment containing the $P_{T7}$- $\Delta N$-$(Gln)_6$-pilE gene was amplified from pHT1447 with a primer set (pSTV28(SmaI-5′)-pCDF-1b-1 and pSTV28(SmaI-3′)-pCDF-1b-2) (S1 Table) and cloned into the SmaI site of pSTV28 (Takara Bio) by In-Fusion Cloning to construct pHT1474 (S3 Table). *E. coli* BL21(DE3) cells harboring pCDF-Pyl-Fx3, pHT1473, and pHT1474 were cultivated in 100 mL MagicMedia containing 1 mM *p*BPa at 30°C overnight. The harvested bacteria were suspended in 1.25 mL PBS and divided into 5 aliquots (0.25 mL/tube), transferred into the wells of a 24-well plate/aliquot, and irradiated with 365 nm UV light on ice for 2 hours. UV-irradiated bacteria were collected again in one tube and disrupted by suspension in 1 mL B-PER (Thermo) containing protease inhibitor cocktail (Nacalai Tesque, Japan), followed by centrifugation at 12,000 rpm at 4°C for 30 min. The MBP-$\Delta N$-PamA K278 (*p*BPa) protein crosslinked to $Gln_6$-$\Delta N$-PilE [as well as uncrosslinked MBP-$\Delta N$-PamA K278 (*p*BPa)] was purified with amylose resin (NEB). The purified fraction was concentrated approximately 70-fold using a VIVASPIN TURBO 15 (MW 50k) Centrifuge Filter unit (Sartorius) to a final volume of approximately 250 μL. An aliquot of the sample was analyzed by 4–12% NuPAGE (Invitrogen), and the bands corresponding to the complex containing MBP-$\Delta N$-PamA K278(*p*BPa) crosslinked to $Gln_6$-$\Delta N$-PilE were excised. In addition, an uncrosslinked band corresponding to MBP-$\Delta N$-PamA K278(*p*BPa) was isolated for a comparative analysis. The band corresponding to the $Gln_6$-$\Delta N$-PilE protein was also excised. The amino acid residue of $Gln_6$-$\Delta N$-PilE crosslinked to $\Delta N$-PamA K278(*p*BPa) was identified by comparing peptides from the three bands as described previously [54].

**Transmission Electron Microscopy (TEM).** Negatively stained samples were prepared for TEM as described previously [55]. Electron microscopy was performed with a Hitachi H-7600 Transmission Electron Microscope.

### Measurement of the ratio of pilus components to PilE in purified pili

Meningococcal pili were prepared as described previously [36]. The relative amounts of PilF--FLAG, PilP-FLAG, PilM-FLAG, PilV, and PilX-FLAG to PilE in purified pili were examined by Western blotting with anti-FLAG mAb (Wako, Japan), anti-PilV [36], or anti-PilE rabbit serum [33] with appropriate dilutions using a quantification program in Fusion Capt 17 in Fusion Solo 7S (M&S Instruments). Results are expressed as means ± SD. The adhered and internalized bacterial numbers and the relative amounts of pilus components to PilE in purified pili were compared using the two-tailed Student's *t*-test, and *P* values of <0.05 were considered significant.

## Results

### Genetic evidence that a hypothetical protein annotated as NMB1345 is important for *N. meningitidis* internalization into HBMEC

During the course of our research on the infectious abilities of the ST-2032 *N. meningitidis* strain mutagenized by STM to HBMEC, we found that a transposon mutant, in which the gene annotated as NMB1345 (renamed as PamA in this study) in the *N. meningitidis* MC58 genome [39] was disrupted and resulted in defective infectious ability in HBMEC (Fig 1A). To confirm the relationship between the mutation and infectious defect in the mutant, we constructed a null mutant (HT1822 *ΔpamA::spc*) and an ectopically complemented strain with the

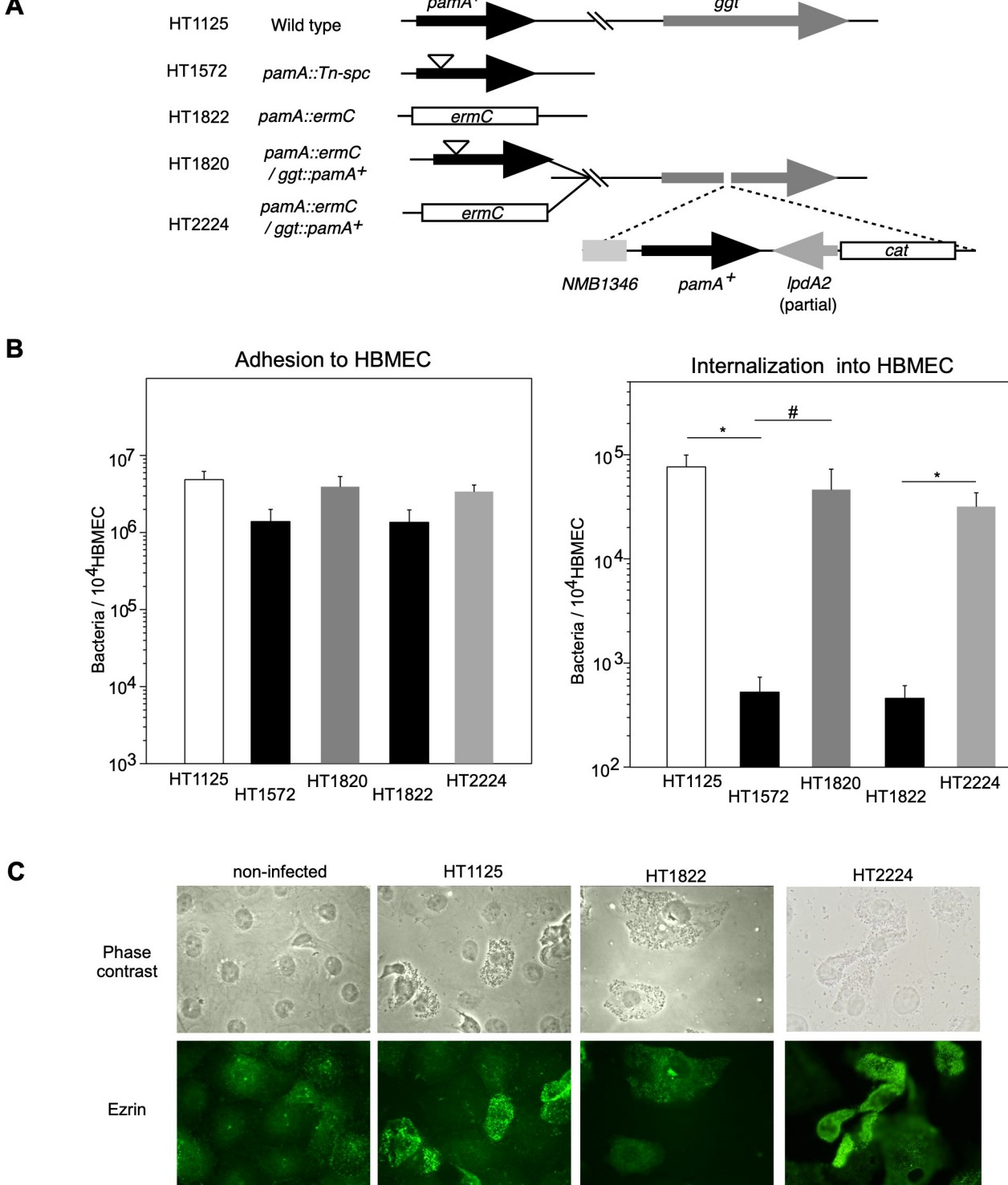

**Fig 1. Characterization of Δ*pamA N. meningitidis* mutant.** (A) Schematic representation of the wild-type, insertion, and deletion mutants in the *pamA* gene and ectopic complementation of the *pamA*+ gene at the *ggt* locus in *N. meningitidis* strains. (B) Effect of Δ*pamA* mutation on *N. meningitidis* infection of HBMEC. The adherence (left) and internalization (right) of *pamA N. meningitidis* mutants to HBMEC, and the effects of complementation of the *pamA*+ gene in the *pamA* deletion mutant on bacterial infection. Each value is the mean ± standard error of the mean (CFU per $10^4$ HBMEC) from at least four experiments. Open, filled, light gray, and dark gray bars indicate the bacterial number of *N. meningitidis* wild-type *pamA*+ (HT1125), *pamA*::*Tn-spc* (HT1572), and Δ*pamA*::*spc* (HT1822), and *pamA*- mutants in which the *pamA*+ gene was ectopically complemented (HT1736 and HT2224), respectively (see S1 Table). *$P$<0.01, #$P$<0.05, significantly different from the *pamA*+ strain or *pamA*-

mutants complemented with the *pamA*[+] gene. (C) Immunofluorescence microscopy showing the accumulation of ezrin beneath *N. meningitidis*-infected HBMEC. The HBMEC monolayer was infected with wild-type *pamA*[+] (middle-left), *ΔpamA* (middle-right), and *ΔpamA/pamA*[+] (right) *N. meningitidis* strains. A non-infected HBMEC monolayer is also shown in the left panels. Bacteria and HBMEC were observed by phase-contrast microscopy (upper panels). Ezrin was immunostained with an anti-ezrin mAb and Alexa Fluor 488-conjugated rabbit anti-mouse IgG (green channel, lower panels).

wild-type *pamA*[+] gene at the *ggt* locus (HT2224) and examined their infectious activities in HBMEC (Fig 1B). While the insertional and null mutants adhered to HBMEC less efficiently than the wild-type strain HT1125, the number of internalized bacteria largely decreased to approximately 1/100 of HT1125 and the defect in internalized bacterial numbers was recovered in HT2224. These results indicate that the mutation in the *pamA* gene affects meningococcal infection in HBMEC, particularly internalization.

We also investigated host cell cytoskeleton rearrangements caused by meningococcal infection by monitoring the localization of ezrin, because the accumulation of ezrin beneath meningococci on HBMEC is required for bacterial internalization into host cells [36, 37] (Fig 1C). While ezrin was widely distributed throughout non-infected cells, it was condensed at the site of bacterial attachment in HBMEC infected with wild-type *N. meningitidis* strain HT1125. In contrast, ezrin condensation was not observed in cells infected with the *ΔpamA N. meningitidis* mutant HT1822, but was detected in HT2224 (*ΔpamA/pamA*[+]). These results are consistent with previous findings, and strongly suggest that PamA participates in the meningococcal-elicited bacteria-induced reorganization of the host cell cytoskeleton upon *N. meningitidis* infection.

## Characterization of the biochemical properties of PamA in *N. meningitidis*

The deduced amino acid sequence of the PamA protein consists of 516 residues (S1 Fig) with a predicted molecular mass of 57 kDa. The hydrophobicity analysis indicated that the PamA protein had a one membrane-spanning region at its N terminus (S1 Fig), suggesting that the PamA protein is localized at the membrane in *N. meningitidis*. To clarify the localization in *N. meningitidis*, cellular proteins were biochemically fractionated using a differential detergent solubilization method [45] and analyzed by Western blotting (Fig 2A). LptA, a meningococcal inner membrane (IM) protein [55], was largely detected in the 4% Triton X-100 soluble (IM-enriched) fraction, supporting the proper fractionation in this experiment. Under the same conditions, the PamA protein was mainly enriched in the IM-rich fraction. This result suggests that PamA is located at the IM in *N. meningitidis*.

We also investigated the membrane topology of the PamA molecule in *N. meningitidis* by constructing *N. meningitidis* mutants, in which the first N-terminal 136 amino acid residues of PamA including the membrane-spanning region, termed *pamA′*, were fused to the *lacZ* or *phoA* gene [56] on the chromosome (Fig 2B)(see Materials and Methods). While the wild-type HT1125 and *pamA′-lacZ N. meningitidis* HT2215 strains (Table 2) did not exhibit β-galactosidase activity, the *pamA′-phoA N. meningitidis* strain HT2217 (Table 2) showed alkaline phosphatase activity (Fig 2C). Since the alkaline phosphatase activity of PhoA was detected only in the periplasmic space [57], these genetic results suggest that the hydrophobic region of the PamA protein faces the periplasmic side. We also biochemically examined the membrane topology, using a Proteinase K digestion experiment with the spheroplasted *N. meningitidis* strain expressing glutathione S-transferase (GST) from an IncQ plasmid (Fig 2D and 2E). Under the condition where the cellular protein GST was mostly protected from Proteinase K digestion (Fig 2D), PamA was largely digested (Fig 2E). Collectively, these results suggest that PamA is localized at the IM and faces the periplasmic space in *N. meningitidis*.

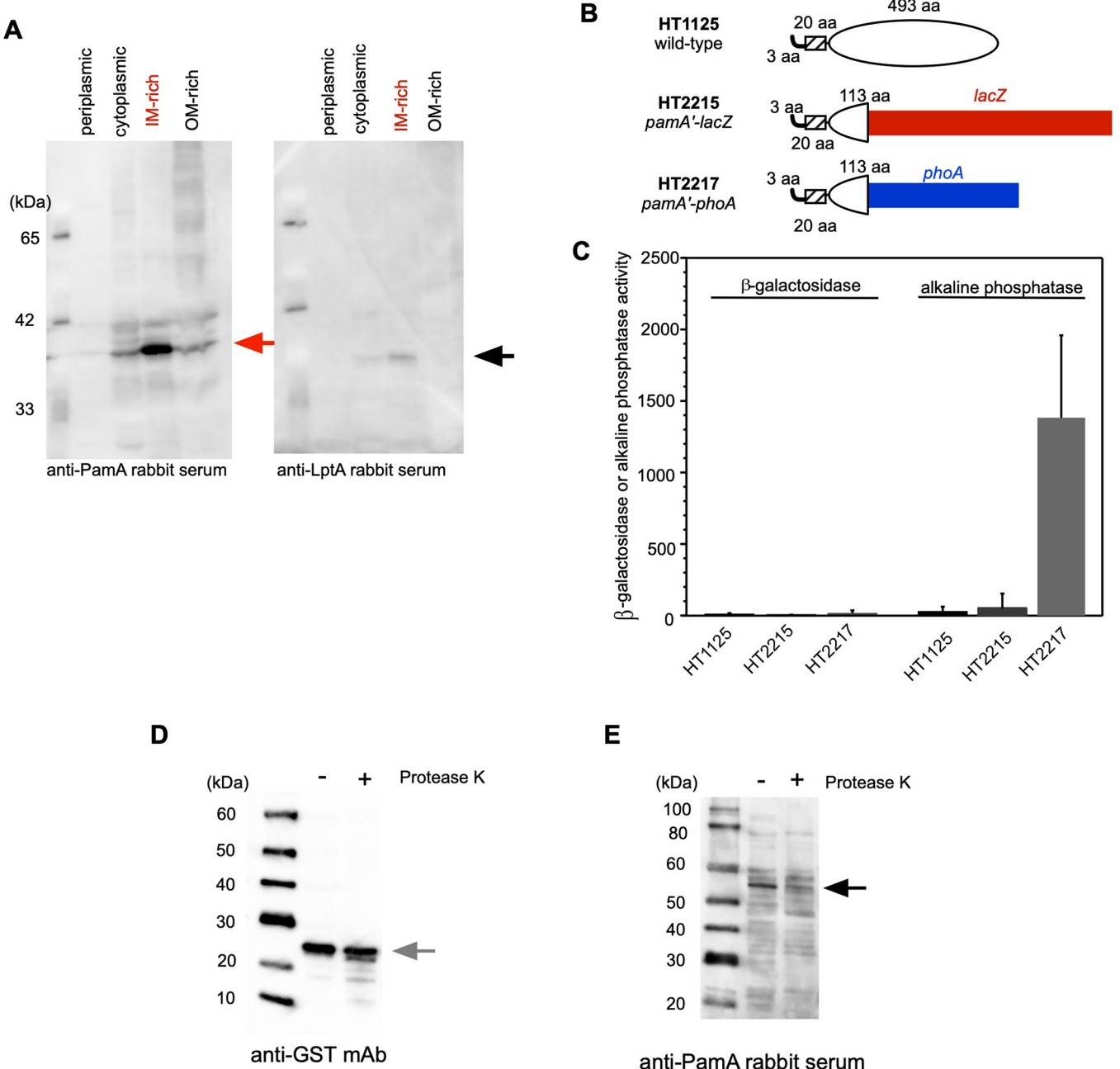

**Fig 2. Characterization of the biochemical properties of the PamA protein in *N. meningitidis*.** (A) Western blot analyses of subcellular fractions of the PamA protein in *N. meningitidis* strain HT1125. A whole-cell lysate periplasmic fraction (P) isolated by CHCl3 shock, cytoplasmic and periplasmic fraction (CP), IM-enriched fraction, and outer membrane (OM)-enriched fraction were prepared by the sonication method from the *N. meningitidis* strain, as described previously [45]. Western blotting with anti-PamA (left) and anti-LptA (right) [55] rabbit antisera. LptA is a control for the proper isolation of the IM-rich fraction. Black and gray arrows show PamA and LptA, respectively. (B) Membrane topology of the PamA protein using the *pamA'-lacZ* and *pamA'-phoA* fused *N. meningitidis* strain. The hatched region consists of the 20 amino acid residues corresponding to the putative membrane-spanning region (S1 Fig). The first 408-bp region of the *pamA* gene was fused to the *lacZ* or *phoA* gene, and the gene was incorporated into the *pamA* locus on the meningococcal chromosome. (C) β-Galactosidase and alkaline phosphatase activities. (D,E) Western blot of spheroplasted *N. meningitidis* strain H44/76 harboring pHT936 (an IncQ plasmid expressing the *gst+* gene, Table 1) followed by a treatment with Proteinase K. Aliquots of the sample were analyzed by Western blotting with anti-GST mAb (D) and anti-PamA rabbit serum (E). Black and gray arrows show PamA and GST, respectively.

## Incorporation of ncAAs into proteins in living *N. meningitidis*

To attempt the application of genetic code expansion to *N. meningitidis* for the incorporation of ncAAs with a photocrosslinker into the PamA protein, we used two mutated *MmPylRS/T* genes to genetically encode ncAAs in *N. meningitidis*. A previous study reported that a modified *MmPylRS* gene for *E. coli*, *MmPylRS*[Y306A/Y384F] [49] is applicable to $N^{\varepsilon}$-(*m*-azidobenzyloxycarbonyl)-L-lysine (*m*AzZLys) (Fig 3A, left), which is a photocrosslinker excited by ultraviolet (UV) irradiation [58], and *MmPylRS*([A302T/N346T/C348T/W417C]) [50] (*Mmp*BPaRS) is optimized for *p*-benzoyl-L-phenylalanine (*p*BPa) (Fig 3A, right), which is also one of the widely used photocrosslinkers [44].

We examined the incorporation of *m*AzZLys and *p*BPa in *N. meningitidis* by biochemically monitoring the suppression of the amber codon at position Glu51 or Phe52 of GST (GST E51amb or GST F52amb) by Western blotting (Fig 3B). The expression of both the *gst E51amb* and *gst F52amb* mutants was suppressed by *p*BPa, while the *gst F52amb* mutant was not suppressed by *m*AzZLys. The suppression efficiency could be considered as approximately 25% of the wild-type GST protein with *p*BPa in the *gst E51amb* mutant, and 37% and 35% with *m*AzZLys and *p*BPa in the *gst F52amb* mutant, respectively. Since the incorporation efficiency of ncAAs is approximately 35–50% in *E. coli* [59], this result suggested that genetic code expansion by the two modified *MmPylRS/T* genes was efficiently conducted in *N. meningitidis* (Fig 3B). These results obtained by biochemical and genetic pyrrolysine-based amber suppression suggest that *m*AzZLys and *p*BPa are incorporated into the GST protein in *N. meningitidis*. The proper incorporation of *p*BPa into a protein in *N. meningitidis*, as determined by an MS analysis, was examined with PamA K278 (*p*BPa)(see a later section, S5 Fig) since the amounts of GST proteins in *N. meningitidis* were too low to purify.

We further examined the UV crosslinking by using whole bacterial cells (Fig 3C). Longer UV irradiation increased the amount of the dimer form of GST, and the majority of GST F52 (*p*BPa) was fixed as the dimer form for 2 hours because GST is a dimeric enzyme [44, 60, 61]. On the other hand, the dimerization of the GST E51amb(*m*AzZLys) and GST E51(*p*BPa) proteins was not observed by photocrosslinking, while GST E51(*m*TmdZLys) dimerized in *E. coli* [62], indicating that the processes involved in UV crosslinking with ncAAs do not always function well in *N. meningitidis*. Collectively, these results demonstrated that physiological transient protein-protein interactions can be detected by crosslinking via the incorporation of photoreactive ncAAs in *N. meningitidis* and showed that the incorporation of ncAAs by pyrrolysine-based amber suppression could be performed in *N. meningitidis*.

## Incorporation of a genetically encoded photocrosslinking amino acid into the PamA protein to identify the accompanying protein in *N. meningitidis*

Since it was not possible to predict the three-dimensional structure of PamA by Phyre2 [63], due to the lack of homology to any other protein identified to date, we were unable to speculate which amino acid residues are exposed on the outside of the PamA molecule in *N. meningitidis*. Therefore, the identities of suitable amino acid residues for the incorporation of photoreactive ncAAs into the PamA protein remained unclear. Thus, in the present study, we focused on the lysine (Lys or K) residue for amber (UAG) substitution because lysine is primarily located on protein surfaces [64, 65]. Forty Lys residues are present in the PamA protein (S1 Fig), and we constructed three *pamA* amber mutants (designated as *pamA K-amb*) at K148 (*pamA K148amb*), K273 (*pamA K273amb*), and K388 (*pamA K388amb*) to estimate the proper position(s) in the PamA protein for ncAAs incorporation. *N. meningitidis* strain H44/76 *ΔpamA* (HT1940) expressing the *MmPylRS[Y306A/Y384F]/T* genes and one of the three *pamA K-amb* genes on an IncQ plasmid was cultivated in the presence of *m*AzZLys, and

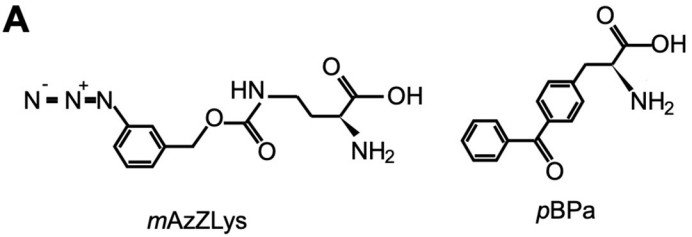

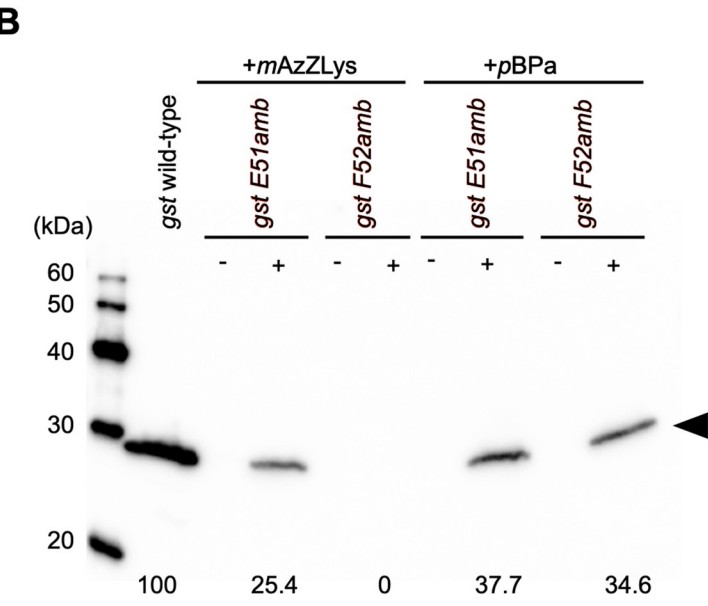

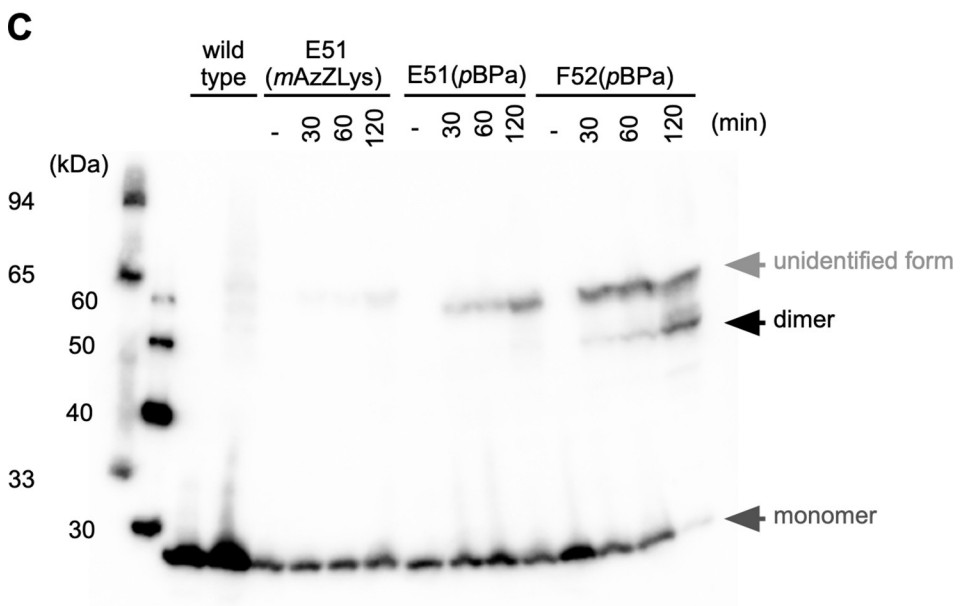

**Fig 3. Incorporation of ncAAs into *N. meningitidis* proteins monitored by Western blotting.** (A) Chemical structures of ncAAs, *m*AzZLys (Left) or *p*BPa (right) used in this study. (B) Estimation of the efficiency of pyrrolysine-based amber suppression with *m*AzZLys or *p*BPa for GST E51amb or GST F52amb in *N. meningitidis*. Bacterial extracts equivalent to an $OD_{600}$ of 0.1 were analyzed by Western blotting with an anti-GST mAb. The horizontal black arrow shows the full-length GST protein. Numbers indicate relative suppression efficiency when wild-type GST expression is defined as 100%. (C) Detection of irradiation- and time-dependent GST dimerization of GST F52 (*p*BPa) in *N. meningitidis*. Bacterial extracts equivalent to an $OD_{600}$ of 0.05, irradiated with UV light for the indicated times, were analyzed by Western blotting with an anti-GST mAb. Dark gray and black arrows show the monomer and dimer forms of the GST protein, respectively. The light gray arrow shows an unspecified form of GST generated by UV irradiation.

whole bacterial extracts were analyzed by Western blotting (S2A Fig). The full length and a sufficient amount of PamA K148amb by incorporation of *m*AzZLys [(PamA K148(*m*AzZLys)], PamA K273(*m*AzZLys), and PamA K388(*m*AzZLys) were only observed in the presence of *m*AzZLys, suggesting that *m*AzZLys is incorporated into the PamA protein by pyrrolysine-based amber suppression in *N. meningitidis*. We also examined UV-irradiated whole bacterial cells by Western blotting, faint bands appeared to correspond to proteins crosslinked to PamA (*m*AzZLys), were detected in the PamA K273(*m*AzZLys) and PamA K388(*m*AzZLys) samples (S2B Fig). The same result was also obtained for *N. meningitidis* strain HT1940 expressing the *MmpBPaRS/T* genes and one of the three *pamA K-amb* genes on an IncQ plasmid, cultivated in the presence of *p*BPa (S3 Fig). These results indicate that the Lys residues located from position Lys148 to the carboxy-terminal end (S1 Fig) were localized on the surface of the PamA molecule in *N. meningitidis*. Moreover, the amber suppression appeared to be more efficiently for PamA K273(*p*BPa) and PamA K388(*p*BPa) with *p*BPa than with *m*AzZLys (Fig 4 and S3 Fig). Thus, the photocrosslinking experiments by pyrrolysine-based amber suppression described hereafter were conducted with the *p*BPa and *MmpBPaRS/T* genes.

To confirm the proper position(s) for the incorporation of *p*BPa into the PamA protein, we constructed 26 *pamA K-amber* mutants, in which each Lys residue located from position Lys148 to the carboxy-terminal end in the PamA protein was replaced with the amber codon (S1 Fig, Table 1 and S4 Table). UV-irradiated *N. meningitidis* strain HT1940 cells expressing the *MmpBPaRS/T* and *pamA K-amber-His₆* genes were analyzed by Western blotting (S3 Fig). Some PamA(*p*BPa) mutants seemed to be crosslinked to some endogenous protein in *N. meningitidis* (shown in red in S3 Fig), and the more detailed analysis revealed that the following six *pamA K-amber* mutants with expression suppressed by *p*BPa [K208(*p*BPa), K278(*p*BPa), K309(*p*BPa), K341(*p*BPa), K382(*p*BPa), and K395(*p*BPa)] appeared to be crosslinked to some endogenous proteins in *N. meningitidis* (shown in red in Fig 4) while K309(*p*BPa) was eliminated. In the present study, we focused only on the proteins crosslinked to PamA K278(*p*BPa), because the three bands with the highest intensities were found in the PamA K278(*p*BPa) sample.

To characterize the 150-, 140-, and 85-kDa protein complexes that contained an endogenous protein crosslinked to PamA K278(*p*BPa), we added a Twin-Strep tag (Strep₂) [66] and His₆ tag at the carboxy-terminal end (S4 Fig) and purified the three corresponding protein complexes from *N. meningitidis* strain HT1940 harboring pHT1388 grown on approximately 200 plates as described in the Materials and Methods. During the purification process, we also obtained the purified PamA K278 (*p*BPa) protein expressed in *N. meningitidis*. The incorporation of *p*BPa at K278amb of PamA-Strep₂ -His₆, was confirmed by MS analyses (S5 Fig). This result demonstrates that an ncAA was selectively incorporated into a protein in *N. meningitidis*.

The three corresponding protein complexes were subjected to a gel-based proteomic analysis using MS (Fig 5A and 5B). A compilation of the results obtained using MASCOT (MATRIX SCIENCE, Japan) suggested several candidates, and we selected those with higher

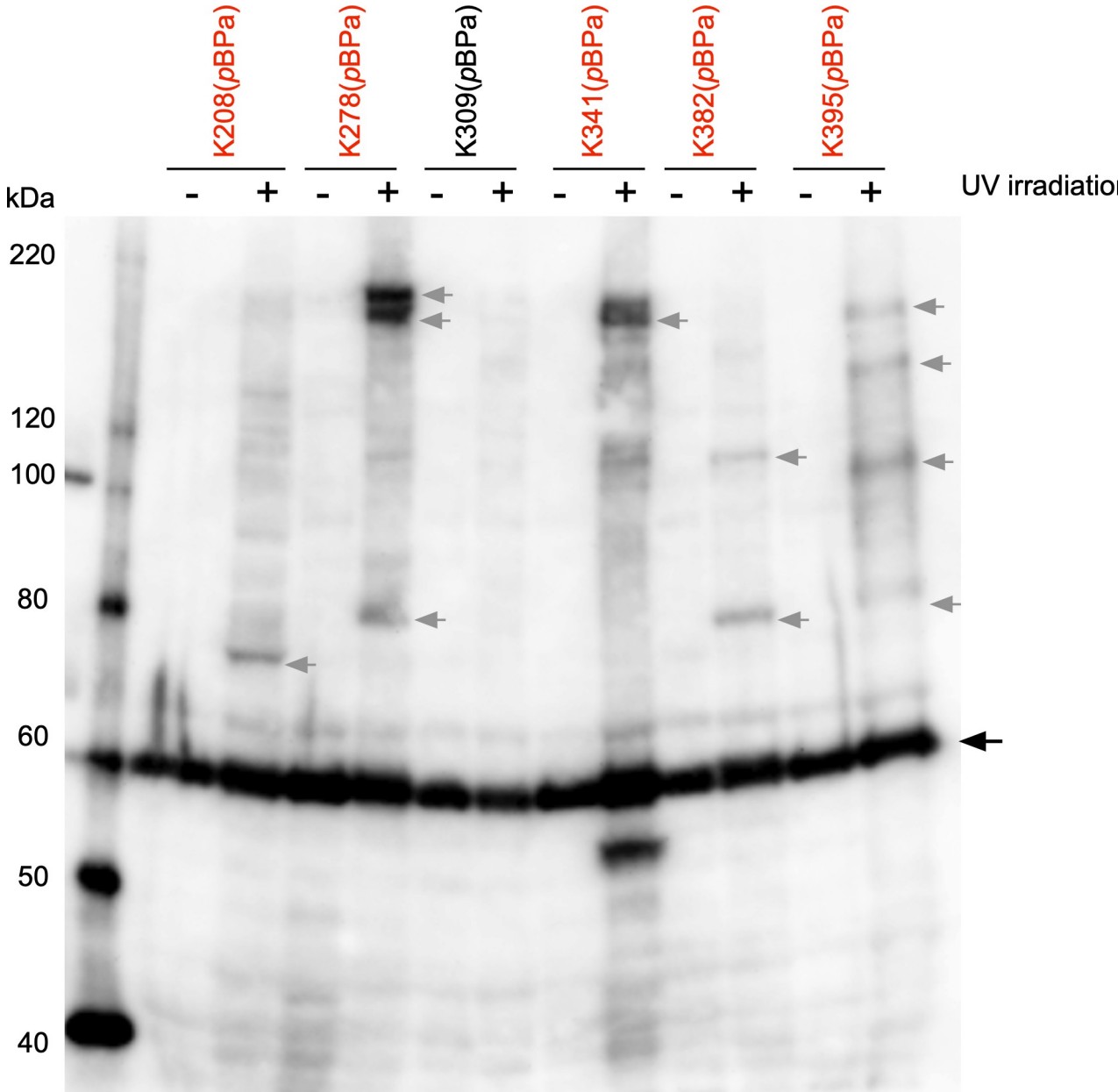

**Fig 4. PamA K208(*p*BPa), PamA K278(*p*BPa), PamA K314(*p*BPa), PamA K382(*p*BPa) and PamA K395(*p*BPa) proteins crosslinked to unidentified endogenous proteins in *N. meningitidis*.** UV crosslinking of Pam K208amb, K278amb, K309amb, K341amb, K395amb, and K382amb mutants expressed by pyrrolysine-based amber suppression with *p*BPa in *ΔpamA N. meningitidis*. Bacteria that grew on one-fourth of a GC agar plate in the presence of 1 mM *p*BPa were crosslinked by UV light irradiation for 2 hours and then the PamA K(*p*BPa)-His$_6$ protein was purified. An aliquot was analyzed by anti-His$_6$ mAb. The black arrow shows the full-length PamA K-amb protein expressed by pyrrolysine-based amber suppression with *p*BPa, and the gray arrow indicates protein complexes crosslinked between PamA K(*p*BPa) and unidentified endogenous proteins in *N. meningitidis*.

scores and numbers of matched peptides for more reliable results in the present study (Table 3). Since the molecular mass of PamA-Strep$_2$-His$_6$ is approximately 62 kDa due to the addition of Strep$_2$-His$_6$ at the carboxy-terminal end (S4 Fig), the bands corresponding to 150- and 140-kDa were presumed to contain counterparts with a molecular mass of approximately 100 kDa. PamA is listed as the top score of the MASTCOT analysis (Table 3) for the 150 and 140 kDa bands, providing support for the presence of this protein in the two bands. On the

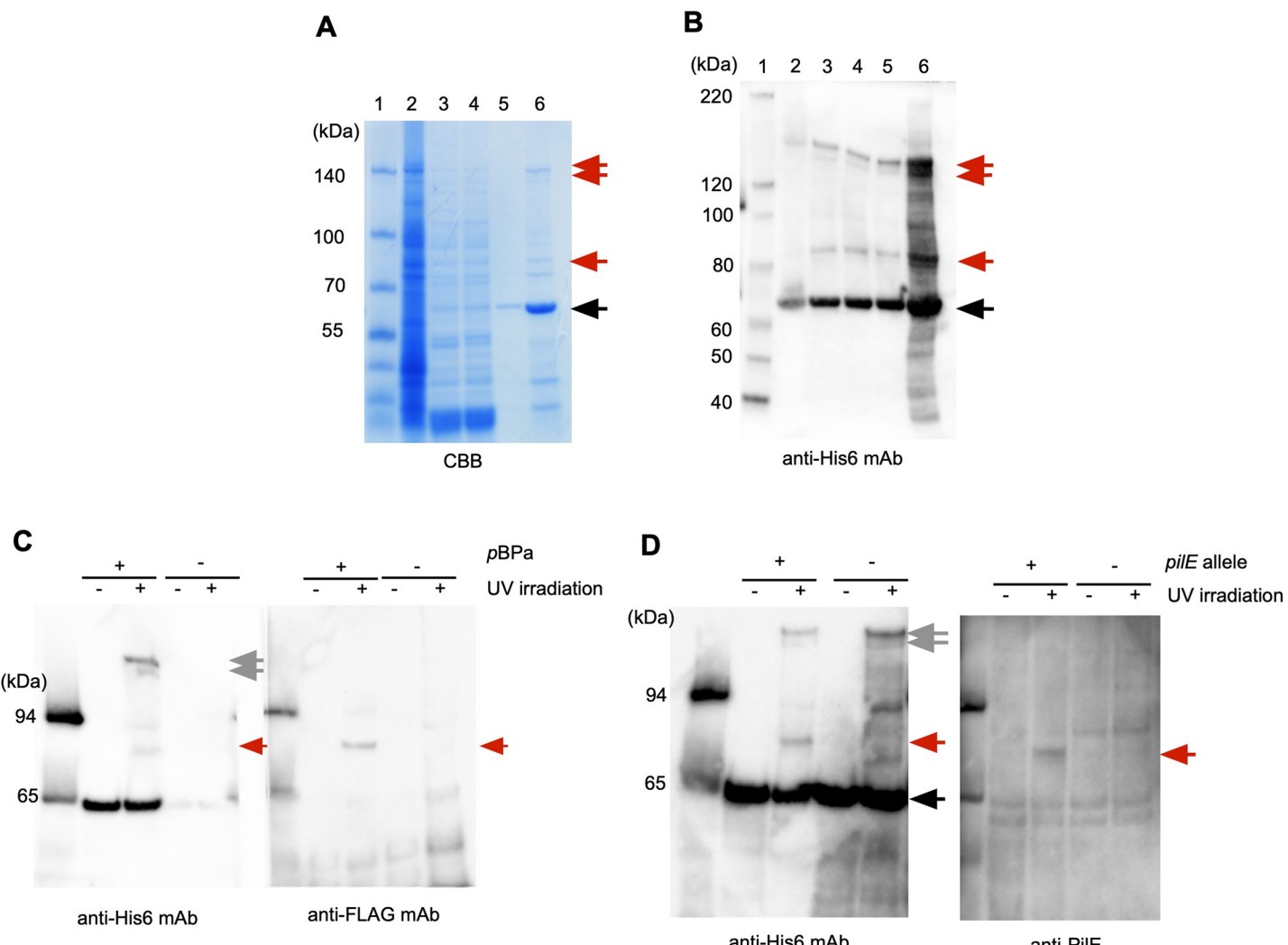

**Fig 5. Purification and identification of an endogenous protein crosslinked to PamA K278(*p*BPa)-Strep₂-His₆.** Proteins in each purification step were analyzed by 4–12% SDS-PAGE and staining with the CBB staining kit (A) and Western blotting with anti-His₆ mAb (B). Lane 1, molecular mass standards; lane 2, *N. meningitidis* crude soluble extract; lane 3, after the Ni-Sepharose column; lane 4, after dialysis with IP Lysis buffer; lane 5, after the Strep-Tactin Sepharose column; lane 6, after concentration with VIVASPIN TURBO 15. Black and red arrows indicate K278(*p*BPa)-Strep₂-His₆ and the unidentified endogenous proteins crosslinked to PamA K278 (*p*BPa)-Strep₂-His₆, respectively. (C) The PilE-FLAG protein was crosslinked to PamA K278(*p*BPa) in *N. meningitidis*. The black arrow shows the full-length PamA K278amb protein expressed by pyrrolysine-based amber suppression with *p*BPa, and the gray arrow indicates the homotrimer (See text) in *N. meningitidis*. The red arrow shows the complex of PilE-FLAG crosslinked to PamA K278(*p*BPa)-Strep₂-His₆ in *N. meningitidis*. +/- indicates the presence or absence of *p*BPa, and irradiation or no irradiation with UV light, respectively. (D) The crosslinked band was reacted with anti-PilE rabbit serum. Black and gray arrows show PamA K278(*p*BPa)-Strep₂-His₆ and the putative homotrimer complex of PamA K278(*p*BPa), respectively, in *N. meningitidis*. The red arrow shows the complex of PilE crosslinked with PamA K278(*p*BPa)-Strep₂-His₆ in *N. meningitidis*. +/- indicates the presence or absence of the *pilE* gene, and irradiation or no irradiation with UV light, respectively.

other hand, we did not identify any candidate with a deduced molecular mass from 80 to 90 kDa by MASCOT (Table 3). To examine the possibility that the observed molecular mass on SDS-PAGE was different from the calculated molecular mass in the database, we constructed *N. meningitidis* mutants that expressed the FLAG-tagged proteins of 5 candidates with estimated molecular masses of 75–160 kDa for 150 kDa, and those of 9 candidates with estimated molecular masses of 48–74 kDa (shown in bold in Table 3). Crosslinked samples from the *N. meningitidis* mutants expressing the FLAG-tagged protein, Mm*p*BPaRS/T and PamA K278 (*p*BPa)-Strep₂-His₆, were examined by Western blotting; however, none of these candidates were crosslinked to PamA K278(*p*BPa) in *N. meningitidis*. Considering these results, the two

**Table 3. Summary of the protein candidates crosslinked to PamA in *N. meningitidis* identified by mass spectrometry.**

1. Protein identified from the photocrosslinking products for a molecular weight around 150kDa

| Accession number | Protein name | *N. meningitidis* strain | Estimated Molecular weight | Number of matched peptides | MASCOT score | Sequence coverage |
|---|---|---|---|---|---|---|
| gi\|325136153 | *hypothetical protein NMBM0579_0886* | *NMBM0579_0886* | *57kDa* | *102* | *1843* | *25%* |
| gi\|325144519 | *hypothetical protein* | *NMBM01240013_0931* | *57kDa* | *99* | *1758* | *24%* |
| gi\|325140435 | *hypothetical protein* | *CU385* | *57kDa* | *93* | *1718* | *24%* |
| gi\|488174415 | *hypothetical protein* | | *57kDa* | *93* | *1920* | *22%* |
| **gi\|485353984** | **DNA-directed RNA polymerase, β subunit** | **73696** | **150kDa** | **24** | **1436** | **22%** |
| gi\|488170470 | *hypothetical protein* | | *57kDa* | *71* | *1394* | *20%* |
| **gi\|6977941** | **App protein** | | **160kDa** | **7** | **425** | **7%** |
| **gi\|915797040** | **adhesin** | | **160kDa** | **6** | **390** | **6%** |
| **gi\|83270238** | **AusI** | | **158kDa** | **4** | **288** | **4%** |
| gi\|389605917 | NAD(P) transhydrogenase subunit alpha | α522 | 54kDa | 2 | 162 | 2% |
| **gi\|488141712** | **carbon starvation protein A** | | **75kDa** | **3** | **146** | **3%** |
| gi\|896381024 | hypothetical protein | | 57kDa | 8 | 119 | 2% |
| gi\|488158877 | hypothetical protein | | 147kDa | 2 | 109 | 2% |
| gi\|254672742 | alanine or glycine: cation symporter, AGCS family | α275 | 45kDa | 2 | 109 | 1% |

2. Protein identified from the photocrosslinking products for a molecular weight around 140kDa.

| Accession number | Protein name | *N. meningitidis* strain | Estimated Molecular weight | Number of matched peptides | MASCOT score | Sequence coverage |
|---|---|---|---|---|---|---|
| gi\|325136153 | *hypothetical protein NMBM0579_0886* | *NMBM0579_0886* | *57kDa* | *80* | *1852* | *25%* |
| gi\|120866758 | *hypothetical protein NMC1281* | *FAM18* | *56kDa* | *69* | *1863* | *23%* |
| gi\|501178984 | DNA-directed RNA polymerase β subunit | | 154kDa | 28 | 1808 | 27% |
| gi\|325140435 | *hypothetical protein NMBCU385_0826* | *CU385* | *57kDa* | *72* | *1803* | *22%* |
| gi\|325144519 | *hypothetical protein* | *NMBM01240013_0931* | *57kDa* | *78* | *1782* | *24%* |
| gi\|488171514 | *hypothetical protein* | | *57kDa* | *70* | *1656* | *23%* |
| gi\|488170470 | *hypothetical protein* | | *57kDa* | *54* | *1267* | *18%* |
| gi\|488186495 | hypothetical protein | | 147kDa | 12 | 800 | 12% |
| gi\|488186095 | immunoglobulin A1 protease | | 121kDa | 9 | 653 | 9% |
| **gi\|488151215** | **peptidylprolyl isomerase** | | **56kDa** | **8** | **500** | **8%** |
| gi\|316985319 | filamentous hemagglutinin family N-terminal domain protein | H44/76 | 261kDa | 8 | 496 | 8% |
| **gi\|488154218** | **heme biosynthesis operon protein HemX** | | **48kDa** | **7** | **490** | **7%** |
| **gi\|488151215** | **peptidylprolyl isomerase** | | **56kDa** | **8** | **465** | **8%** |
| **gi\|488146114** | **D-lactate dehydrogenase** | | **64kDa** | **5** | **363** | **5%** |
| **gi\|389605276** | **UPF0141 inner membrane protein yhjW** | | **62kDa** | **6** | **351** | **5%** |
| gi\|488170091 | peptidase | | 200kDa | 3 | 202 | 3% |
| **gi\|120866107** | **putative protein-export membrane protein** | **FAM18** | **66kDa** | **3** | **176** | **3%** |
| **gi\|732853** | **IgA1 protease** | | **54kDa** | **3** | **168** | **3%** |
| **gi\|488141712** | **carbon starvation protein A** | | **74kDa** | **3** | **165** | **3%** |
| gi\|316985875 | 4Fe-4S binding domain protein | | 146kDa | 3 | 150 | 3% |
| **gi\|389605917** | **NAD(P) transhydrogenase subunit α** | | **54kDa** | **2** | **140** | **2%** |
| **gi\|488141355** | **arginine decarboxylase** | | **71kDa** | **2** | **129** | **2%** |
| gi\|349520 | pilus structural subunit, partial | | 17kDa | 2 | 128 | 2% |
| **gi\|488143525** | **protein translocase component YidC** | | **60kDa** | **2** | **125** | **2%** |
| **gi\|254670993** | **1-deoxy-D-xylulose 5-phosphate synthase** | | **69kDa** | **2** | **114** | **2%** |
| **gi\|304337210** | **methionine-R-sulfoxide reductase** | | **59kDa** | **2** | **112** | **2%** |
| gi\|4838369 | NatD | | 52kDa | 2 | 109 | 2% |

*(Continued)*

**Table 3.** (Continued)

3. Protein identified from the photocrosslinking products for a molecular weight around 85kDa.

| Accession number | Protein name | *N. meningitidis* strain | Estimated Molecular weight | Number of matched peptides | MASCOT score | Sequence coverage |
|---|---|---|---|---|---|---|
| gi\|2460281 | outer membrane protein Omp85 | | 88kDa | 63 | 2799 | 41% |
| gi\|488170971 | membrane protein | | 88kDa | 58 | 2599 | 38% |
| *gi\|325136153* | *hypothetical protein NMBM0579_0886* | | *57kDa* | *63* | *1774* | *24%* |
| *gi\|325144519* | *hypothetical protein NMBM01240013_0931* | | *57kDa* | *60* | *1692* | *23%* |
| *gi\|488171514* | *hypothetical protein* | | *57kDa* | *56* | *1619* | *22%* |
| *gi\|488174415* | *hypothetical protein* | | *57kDa* | *56* | *1577* | *22%* |
| gi\|325141117 | LPS-assembly protein LptD | CU385 | 87kDa | 21 | 1440 | 19% |
| gi\|488147360 | elongation factor G | | 77kDa | 27 | 1398 | 20% |
| gi\|325141117 | putative organic solvent tolerance protein | CU385 | 87kDa | 20 | 1149 | 18% |
| gi\|409107079 | type IV pilus biogenesis and competence protein PilQ | | 80kDa | 17 | 973 | 15% |
| gi\|209363328 | hemoglobin receptor, partial | | 87kDa | 16 | 909 | 14% |
| gi\|488154195 | ATP-dependent Clp protease ATP-binding subunit ClpA | | 85kDa | 21 | 905 | 16% |
| gi\|30017077 | TonB-dependent siderophore receptor FetA | | 73kDa | 14 | 793 | 13% |
| gi\|120866913 | RNA polymerase sigma factor | | 75kDa | 11 | 616 | 10% |
| gi\|325142240 | 2-oxoglutarate dehydrogenase E1 component | | 102kDa | 10 | 496 | 8% |
| gi\|488150134 | GNAT family N-acetyltransferase | | 89kDa | 7 | 421 | 7% |
| <u>gi\|496712676</u> | <u>fimbrial protein</u> | | <u>18kDa</u> | <u>9</u> | <u>307</u> | <u>4%</u> |
| gi\|645213765 | peptidase | | 88kDa | 5 | 295 | 5% |
| **gi\|402319479** | **hypothetical protein NMEN93004_1215** | | **18kDa** | **6** | **294** | **5%** |
| gi\|488144915 | phosphoenolpyruvate synthase | | 87kDa | 5 | 293 | 5% |
| gi\|254673965 | putative efflux system transmembrane protein | | 113kDa | 5 | 249 | 4% |
| gi\|496706569 | transferrin-binding protein 2 | | 74kDa | 5 | 239 | 4% |
| gi\|896272027 | pilus assembly protein PilQ | | 75kDa | 5 | 222 | 4% |
| **gi\|45245** | **periplasmic iron-binding protein** | | **34kDa** | **4** | **204** | **4%** |
| gi\|121052755 | GTP pyrophosphokinase | | 86kDa | 3 | 199 | 3% |

Proteins shown in italics corresponds to the PamA protein itself.

Proteins shown in bold indicates the candidates examined for crosslinking to PamA K278(*p*BPa) in *N. meningitidis* by constructing the FLAG-tagged candidate mutants, but the crosslinking could not be detected.

The protein shown underlined is PilE crosslinked to PamA K278(*p*BPa) in *N. meningitidis*, which was confirmed in this study.

proteins presented by the 150- and 140-kDa bands with the highest intensities on Western blotting apparently correspond to the homo-trimer of the PamA protein (see Discussion and S12 Fig).

We then characterized the 85-kDa band, which could contain an approximately 20 kDa endogenous protein crosslinked to PamA K278(*p*BPa). LC-MS/MS followed by MASCOT analyses revealed two possible candidates, a fimbrial protein (17 kDa) and the hypothetical protein NMEN93004_1215 (18 kDa) (Table 3). The BLAST homology search indicated that while we could not find any protein corresponding to NMEN93004_1215 in the databases, the fimbrial protein (accession number gi|496712676) was 95% identical to the protein NMBH4476_0018 annotated as "type IV pilus assembly protein PilA" in *N. meningitidis* strain H44/76 with the highest E-value (Table 3). Further computational analyses revealed that the

"PilA protein" annotated in GenBank was 96% identical to PilE, a major component of meningococcal pili (reviewed in [17]). To confirm that the 85-kDa complex was composed of PilE and K278(*p*BPa)-Strep$_2$-His$_6$, we constructed the *pilE-FLAG N. meningitidis* strain HT2095 and examined the crosslinked samples by Western blotting with an anti-FLAG mAb (Fig 5C). The analysis revealed that PilE-FLAG was present in the 85-kDa band, with dependence on *p*BPa and UV irradiation (Fig 5C). We further examined crosslinked samples from the wild-type (HT1940 harboring pHT1388) and *pilE*-insertional mutant (HT2014 harboring pHT1388) by Western blotting with anti-PilE rabbit serum [33] (Fig 5D). The 85-kDa band was also observed with anti-PilE rabbit serum in the *pilE*$^+$ genetic background only with *p*BPa and UV irradiation. Considering these results, we conclude that the 85-kDa band contains the PilE protein, indicating that PamA interacts with PilE in *N. meningitidis*.

## Mapping the amino acid residue in PilE that crosslinks to the PamA protein

To further clarify the interaction between PamA and PilE, we attempted to identify the amino acid residue of PilE that crosslinks with PamA K278(*p*BPa). Since mapping was difficult and tedious [42], particularly for the endogenous proteins, the examination was conducted with recombinant proteins in *E. coli*.

We examined the interaction of two truncated recombinants, ΔN-PamA K278amb fused to maltose-binding protein (MBP) (MBP-ΔN-PamA K278amb) and ΔN-PilE with six glutamines (Gln$_6$) in *E. coli* (S6 Fig). The crosslinked complex between MBP-ΔN-PamA K278(*p*BPa) and Gln$_6$-ΔN-PilE was purified from *E. coli* and investigated using a gel-based proteomic analysis with LC-MS/MS (S7–S9 Figs) [42]. The peptides containing *p*BPa (IEVGK*[*p*BPa]LAFSTK), corresponding to aa 639 to 650 of MBP-ΔN-PamA K278(*p*BPa) (corresponding to aa 274 to 284 of native PamA), from crosslinked and uncrosslinked samples were analyzed by MS/MS. As shown in S9 Fig, the sequence was read from the annotated b and y ions of a crosslinked peptide until isoleucine (Ile) at position 19. This result indicated that the region around Ile19 in Gln$_6$-ΔN -PilE is the site crosslinked to K278(*p*BPa) in MBP-ΔN-PamA.

## Genetic elucidation of the interaction between PamA and PilE in *N. meningitidis* and its important for meningococcal internalization into HBMEC

Ile at position 19 in Gln$_6$-ΔN-PilE corresponded to position 12 in native PilE in *N. meningitidis* (S10 Fig) because prepilin is processed by the prepilin peptidase PilD during translocation at the IM in *N. meningitidis* [36, 67]. Multiple alignments showed that the N-terminal region of PilE around Ile12 is highly conserved among neisserial species, including *N. gonorrhoeae* and *N. lactamica*, implicating the importance of the region in PilE (S10A Fig) [68].

To further examine the functional interaction between PamA and the region containing the PilE I12 residue in *N. meningitidis*, the meningococcal *pilE I12A-cat* mutant HT2167 was constructed (Table 1), and we monitored its infectious ability and the accompanying host cell's cytoskeleton rearrangement by the localization of ezrin. The *pilE I12A N. meningitidis* mutant was less efficiently internalized into HBMEC, while the adhesion remained unchanged (Fig 6A), and ezrin did not accumulate beneath the *pilE I12A-cat* mutant (Fig 6B). Western blotting with UV-irradiated samples purified from HT2167 expressing the *MmpBPaRS/T* and *pamA K278amb* genes also confirmed that the interaction between PilE and PamA K278(*p*BPa) is markedly reduced by the *pilE I12A* mutation (Fig 6C), providing genetic evidence for the interaction between PamA and PilE in *N. meningitidis*. Furthermore, since the phenotypes appeared to be mostly identical to that of the Δ*pamA N. meningitidis* mutant (Figs 1 and 6A), these results indicated that PamA and PilE interact functionally with each other in *N. meningitidis* and that this interaction is important for meningococcal infection in HBMEC.

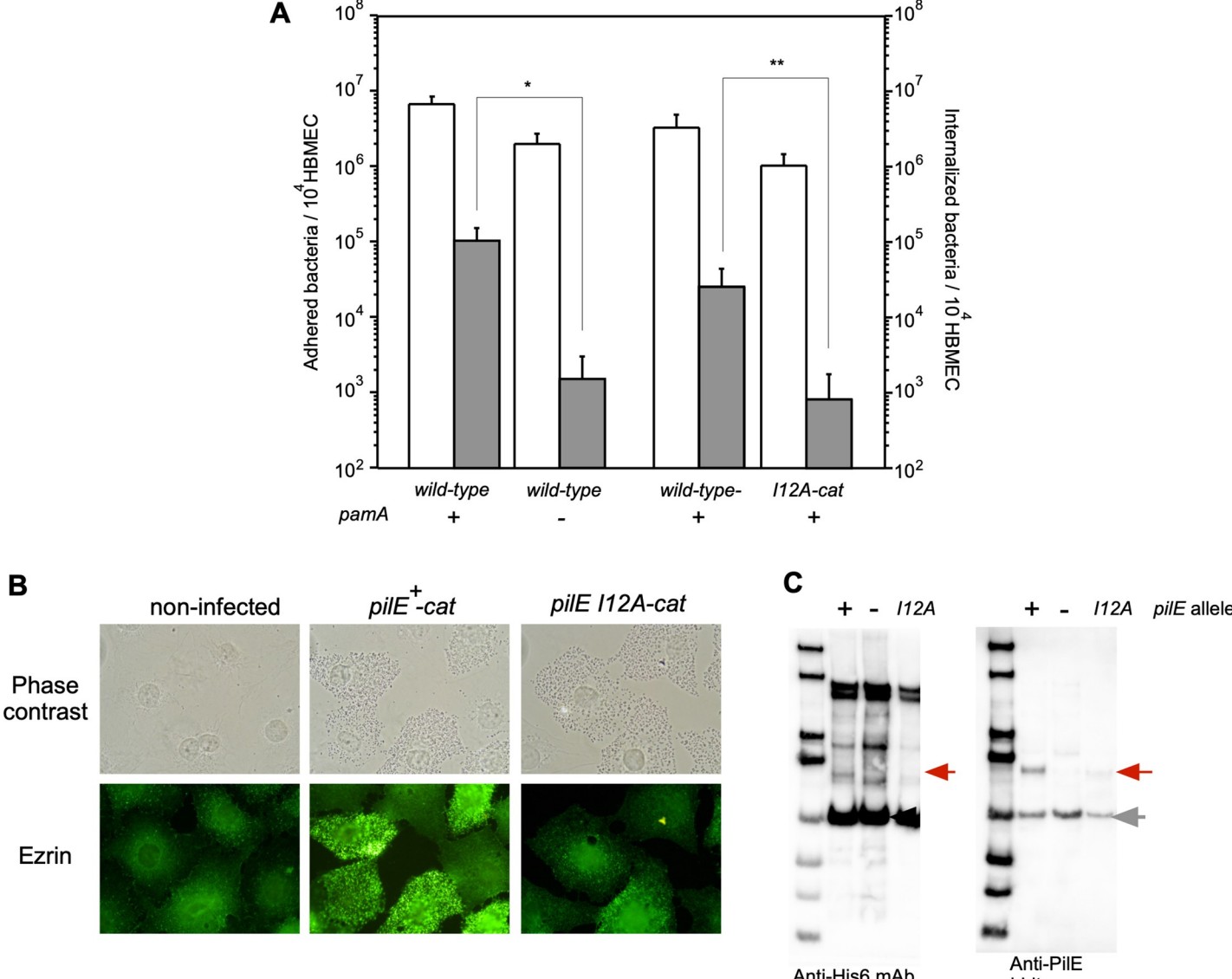

**Fig 6. The *pilE I12A N. meningitidis* mutant also showed reduced internalization into HBMEC, the induction of cytoskeleton rearrangements, and interactions with PamA *in N. meningitidis*.** (A) A I12A mutation in the *pilE* gene affects *N. meningitidis* internalization into HBMEC. Adherence (white) and internalization (gray) of *N. meningitidis* strains in HBMEC. Each value is the mean ± standard error of the mean (CFU per $10^4$ HBMEC) from at least four experiments. The bars indicate the bacterial numbers of *N. meningitidis* wild-type *pamA*[+] (HT1125, left), *ΔpamA::spc* (HT1822, middle-left), *pilE*[+]-*cat* (HT1744 middle-right), and *pilE I12A-cat* (HT2167, right), respectively (see S1 Table). *$P<0.01$, **$P<0.02$, significantly different from the *pamA*[+] strain or *pilE*[+]-*cat* strain. (B) Immunofluorescence microscopy showing the accumulation of ezrin beneath *N. meningitidis* strains. The HBMEC monolayer was infected with wild-type *pilE*[+]-*cat* (middle) and *pilE I12A-cat* (right) *N. meningitidis* strains. A non-infected HBMEC monolayer is also shown in the left panels. Bacteria and HBMEC were observed by phase-contrast microscopy (upper panels). Ezrin was immunostained with anti-ezrin mAb and Alexa Fluor 488-conjugated rabbit anti-mouse IgG (green channel, lower panels). (C) PilE I12A was crosslinked to PamA K278(*p*BPa) less efficiently than wild-type PilE in *N. meningitidis*. Black and red arrows show the crosslinked complex between PamA K278(*p*BPa) and PilE, and the gray arrow indicates an unidentified band that crossreacted with anti-PilE rabbit serum. +/- indicates the presence or absence of the *pilE* gene, respectively.

## Characterization of pili in the *ΔpamA N. meningitidis* mutant

The results described above suggested that PamA affects the functions of meningococcal pili, resulting in a negative impact on meningococcal infection in HBMEC. To further clarify the effects of PamA on pili, we investigated the function of pili in *N. meningitidis*. Electron

microscopy with negative staining showed that the *ΔpamA N. meningitidis* mutant was similarly piliated to the wild-type strain HT1125 (S11A Fig, left and middle-left). Meningococcal pili are known to play a role not only in the infection of host cells [17, 69], but also in meningococcal natural competence, switching motility, and aggregation [70, 71]. An examination of pilus function in natural competence and aggregation revealed that the pili of the *ΔpamA N. meningitidis* mutant retain similar functions to those of the wild-type *pilE N. meningitidis* strain (S11B and S11C Fig). These results suggest that PamA does not participate in pilus formation itself or in functions related to natural competence and aggregation. It is important to note that the *pilE I12A-cat N. meningitidis* mutant is also as functional as the wild-type *pilE-cat N. meningitidis* strain while its piliation number its apparently smaller possibly due to translational fusion with the *cat* gene (S11 Fig and Fig 6).

We examined the effects of PamA on meningococcal pili in more detail. Although the main component of the meningococcal pilus is PilE, its formation requires more than 20 proteins [72–74]. To investigate the effects of the *pamA* mutation on pili, we initially assessed the pilus protein levels in HT1125 and HT1822 using LC-MS/MS with Tandem Mass Tag (TMT) labeling [38] (S5 Table), and found that the amounts of some pilus components appeared to differ between HT1125 and HT1822. However, accurate comparisons of the protein amounts, particularly that for PilE, by TMT are methodologically difficult because *N. meningitidis* has 8 non-expressed *pilS* genes [75], according to the deduced amino acids recorded as Protein in the database. Therefore, we directly monitored the contents of pilus components, particularly for three minor pilins: PilX for aggregation in and adherence to human cultured cells [71, 76, 77], PilV to trigger plasma membrane reorganization [36, 71, 78], and ComP for natural competence [79]. PilF, PilM, and PilP, located on the meningococcal inner membrane for pilus formation [8, 67, 69, 80, 81], were also monitored. The relative amounts of the components to the PilE subunit in purified pili were measured by Western blotting followed by quantification (Fig 7). The relative ratios of PilF, PilP, and PilM to PilE did not significantly differ between the *ΔpamA* and wild-type *N. meningitidis* strains. On the other hand, in *ΔpamA* pili the relative amount of PilV increased, whereas that of PilX decreased (Fig 7A). It is important to note that the amount of ComP was too low in the whole cell extract or purified pili to detect by WB, even when the *comP-FLAG N. meningitidis* strain or anti-ComP rabbit antiserum was used (data not shown). We also confirmed that the expressions of PilE, PilV, and PilX-FLAG were not affected by the *ΔpamA* mutation (Fig 7B). Considering these results, the *pamA* mutation may cause unusual proportions of PilV and PilX to PilE in meningococcal pili, probably by improper sorting of PilE during pili formation, which might affect meningococcal infection in human cultured cells.

## Discussion

While progressive genome sequencing has recently provided novel insights into *N. meningitidis*, more than 50% of the annotated ORFs remain as "hypothetical proteins", and their biological functions are currently unclear. This may be due to an incomplete understanding of meningococcal pathogenesis, which is largely attributable to the lack of analytical methods for *N. meningitidis* [34]. While the results of our genetic analyses indicated that the hypothetical protein NMB1345 (renamed PamA in the present study) in *N. meningitidis* plays an important role in meningococcal internalization into human cultured cells, the novel methodologies currently available for meningococci did not provide any insights into the function of PamA in *N. meningitidis*. Therefore, the incorporation of photoreactive ncAAs by pyrrolysine-based amber suppression was applied to a meningococcal protein with an unknown function, which implicated its physiological function in *N. meningitidis*.

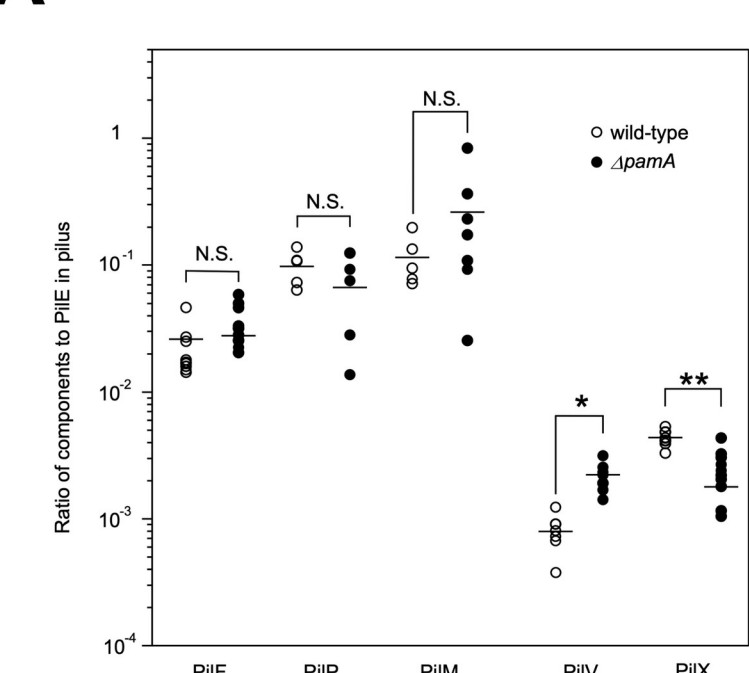

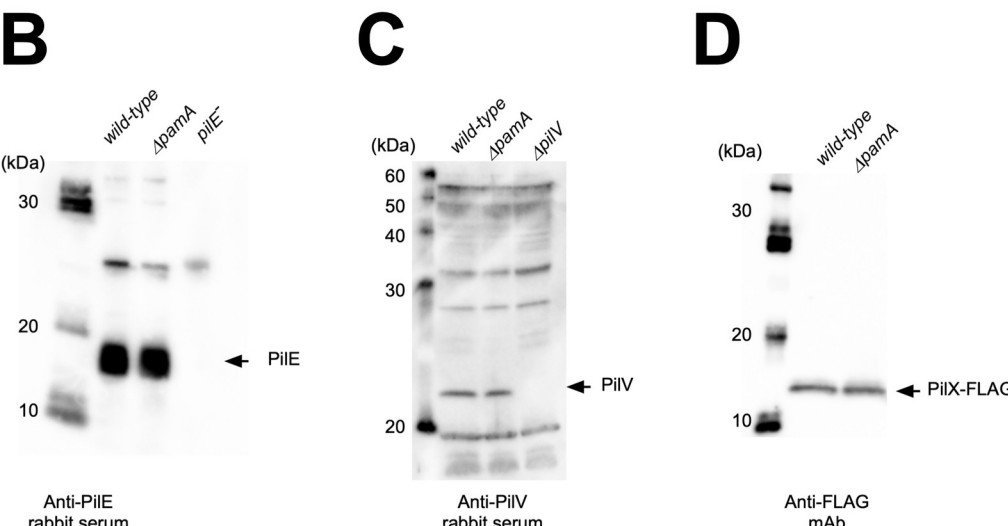

**Fig 7. Disruption of the interaction between PamA and PilE leads to abnormal proportions of PilV and PilX relative to PilE in meningococcal pili.** (A) Ratio of pilus components to PilE in purified pili. The relative amounts of PilF-FLAG (HT2218 and HT2219), PilP-FLAG (HT2136 and HT2137), PilM-HA (HT2132 and HT2133), PilV (HT1125 and HT1822), and PilX-FLAG (HT2211 and HT2212) to PilE in purified pili were measured by Western blotting with appropriately diluted samples followed by quantification of the band intensity using the 30-kDa band in the molecular marker as an internal golden standard. The values obtained were divided by the value of PilE in the same sample. The significance of differences was examined by the t-test. (B, C, D) The *pamA* mutation does not affect the expression of the PilE (B), PilV (C), or PilX-FLAG (D) protein in *N. meningitidis*. Bacterial extracts equivalent to $OD_{600}$ value of 0.0025 (for PilE) or 0.05 (for PilV and PilX-FLAG) were analyzed by Western blotting with anti-PilE and anti-PilV rabbit sera, and an anti-FLAG mAb, respectively. Black arrows indicate PilE, PilV, and PilX-FLAG, respectively.

The incorporation of ncAAs by pyrrolysine-based amber suppression has been used for site-specific incorporation into target proteins *in vivo* in a broad range of species from *E. coli* to eukaryotic systems (reviewed in [41]). However, regarding pathogenic bacteria, only a few studies have successfully incorporated ncAAs into EPEC [82], *Shigella flexinelli* [82], *Salmonella typhimurium* [83], *M. tuberculosis* [84], and *B. cereus* [85]. Moreover, these bacteria were only subjected to examinations of the incorporation of ncAAs into a well-characterized protein (e.g. GST or GFP), with a low molecular mass (approx. 10 kDa) and a known function [86], or a peptide [85], produced from expression vectors. Moreover, X-ray structure information for target proteins will also be advantageous for photoreactive ncAAs incorporation because the proper positions to incorporate ncAAs may be more easily predicted. Even when the X-ray structure of a target protein is unknown, a lower molecular mass is more advantageous because it is easier to scan the optimal site for ncAA incorporation in the target protein. Therefore, genetic photocrosslinker incorporation has not yet been applied to analyze a protein with unknown function or three-dimensional structure is unknown. Furthermore, some disadvantages in the application of photocrosslinking ncAAs incorporation combined with MS to *Neisseria* species are the very few expression vectors for neisseriae [33, 34] [87] and the limited culture methods in liquid medium due to autolysis [88]. To the best of our knowledge, this is the first study to demonstrate the incorporation of ncAAs into *N. meningitidis*. Moreover, after observing that ncAA-incorporated GST proteins are crosslinked as the dimer form in *N. meningitidis* (Fig 3E), we incorporated *p*BPa into the PamA protein in *N. meningitidis* (S5 Fig) and found that this meningococcal hypothetical protein with no physiological and structural information is closely associated with PilE in *N. meningitidis*.

Our strategy of incorporating a genetic photocrosslinker coupled with MS in *N. meningitidis* revealed that the meningococcal hypothetical protein PamA is associated with PilE in *N. meningitidis*. Further physicochemical analyses with PamA and PilE recombinants in *E. coli* identified the crosslinked regions between PamA K278(*p*BPa) PilE (S7–S9 Figs), and genetic analyses using the *pilE I12A N. meningitidis* mutant suggested that the interaction between PamA and PilE in *N. meningitidis* is functionally important for meningococcal pilus formation and infection to HBMEC (Fig 6). While further studies are needed, the PamA interaction with the N-terminal region of PilE around the Ile12 residue (S10 Fig) may be important to maintain the function of meningococcal pili by PamA *in N. meningitidis*.

The function of meningococcal PamA has not been clearly discerned in this study. The present results showed that PamA is not required for pilus formation itself (S11A Fig) but participates to keep optimal proportions of PilV and PilX to PilE in meningococcal pili (Fig 7). This result does not seem to contradict previous findings showing that the proper ratio of PilV and PilX to PilE in pili leads to pilus conformational changes that trigger plasma membrane reorganization [36, 71, 78, 76, 77]. However, we could not eliminate the possibility that more essential dysregulation(s) other than the imbalance of PilV and PilX to PilE in meningococcal pili by the *pamA* mutant were not identified in this study. We focused on three bands crosslinked with PamA K278(*p*BPa) with the highest intensity among the five PamA K-amb (*p*BPa) mutants, while the interaction between PamA and PilE found was identified from a band with the lowest intensity (Fig 4). On the other hand, the two protein complexes with the highest intensities (approximately 150 and 140 kDa) crosslinked to PamA K278(*p*BPa) were not identified in any protein other than PamA itself by peptide mass fingerprinting (Table 3). Although further analyses are needed, the two bands appeared to correspond to the homotrimer of PamA because intramolecular crosslinking was also detected in the MBP-ΔN-PamA K278 (*p*BPa) recombinant protein in *E. coli* (S12 Fig). The endogenous proteins crosslinked to the PamA K208(*p*BPa), PamA K341(*p*BPa), PamA K382(*p*BPa), and PamA K395(*p*BPa) mutants

have yet to be identified (Fig 4) and their characterization will provide more information about the novel function of PamA in *N. meningitidis*.

Surprisingly, unidentified genes involved in pilus function are still being discovered, although more than 20 proteins involved in pilus synthesis and function have already been identified by genetic [72–74] and WGS [32, 39] analyses. Therefore, our strategy using genetic screening by STM combined with genetically-encoded photocrosslinking amino acid incorporation and an MS analysis, could be a powerful method to identify new pathogenic factors in *N. meningitidis*. BLAST searches revealed *pamA* gene homologues in only three neisserial species: *N. meningitidis*, *N. gonorrhoeae*, and *N. lactamica* (S13 Fig). *N. meningitidis* and *N. gonorrhoeae* are closely related bacterial pathogens that share many common genes with a high degree of sequence identity (typically greater than 95%), and DNA relatedness studies clustered commensal *N. lactamica* with these two pathogenic *Neisseria spp.* [89]. Since the sequences of the *pamA* genes appeared to be more divergent among the three species than within *N. meningitidis* (S13 Fig), the *pamA* gene may have been acquired at a later stage of evolution by the three neisserial species, and the restricted possession of the *pamA* gene in the three neisserial species may reflect the physiological function of PamA.

While further studies are needed to clarify the functions of PamA in *Neisseria*, it is important to note that the application of new methodology to *N. meningitidis* provided novel insights into the physiological function of a meningococcal hypothetical protein with no obvious information such as its predicted biological function or three-dimensional structure. Our established system for ncAA incorporation by pyrrolysine-based amber suppression into *N. meningitidis* will facilitate a broad range of *N. meningitidis* studies, particularly the mapping of protein-protein interaction networks or characterization of the functions of hypothetical proteins *in vivo*. Nevertheless, as shown in the present study, *in vivo* research involving the combination of STM and photoreactive ncAAs incorporation by pyrrolysine-based amber suppression with an MS analysis will reveal meningococcal virulence factors that are difficult to identify using most currently available methods. The ncAA incorporation system described in the present study may become one of the strongest driving advancements for studies on meningococcal biology, pathogenesis, and vaccine development.

## Supporting information

**S1 Fig. Nucleotide sequence and deduced amino acid sequence of the *pamA* gene in *N. meningitidis* HT1125 strain.** (GenBank Accession No LC511747). Lysine (K) residues are shown in red and the K residues replaced with amber codons in this study are indicated in bold red. Positions of K residues are shown as blue numbers. The grey bar indicates the putative hydrophobic region identified by SOSUI [90], which region were deleted to construct the PamA recombinants in *E. coli* in this study.
(PDF)

**S2 Fig. Photocrosslinking of PamA K148(*m*AzZLys), K273(*m*AzZLys), and K388(*m*Az-ZLys) in *N. meningitidis*.** (A) Detection of PamA K148amb, K273 and K388amb expressed by pyrrolysine-based amber suppression with *m*AzZLys in *N. meningitidis*. Bacteria grown on one-fourth of a GC agar plate in the presence or absence of 0.3 mM *m*AzZLys and extracts equivalent to $OD_{600}$ of 0.1 were analyzed by Western blotting with anti-PamA rabbit serum. +/- indicates the presence or absence of *m*AzZLys. The black arrow shows the full-length PamA protein. (B) Detection of complexes containing PamA K(*m*AzZLys) crosslinked to an endogenous protein in *N. meningitidis*. Bacteria grown on one-fourth of a GC agar plate in the presence of 0.3 mM *m*AzZLys were crosslinked by UV light irradiation, and mixed with SDS buffer. Aliquots were analyzed by Western blotting with an anti-His$_6$ mAb. +/- indicates

presence or absence of *m*AzZLys (upper), treatment or nontreatment with UV irradiation (lower). Black and grey arrows show the full-length PamA protein and putative complexes corresponding to PamA K(*m*AzZLys) crosslinked to an endogenous protein, respectively.
(PDF)

**S3 Fig. PamA K278(*p*BPa) protein crosslinks with endogenous proteins in *N. meningitidis*.** UV crosslinking of 27 PamA K-amb mutants expressed by pyrrolysine-based amber suppression with *p*BPa in *ΔpamA N. meningitidis*. The black arrow shows the full-length PamA K-amb protein expressed by pyrrolysine-based amber suppression with *p*BPa. PamA K-amb mutants shown in red were subjected to a more detailed analysis in Fig 4.
(PDF)

**S4 Fig. Schematic structure of the C-terminal region of PamA K278-Strep$_2$-His$_6$.** (A) Amino acid sequence of the C-terminal region of PamA K278-Strep$_2$-His$_6$. (B) Nucleotide sequence of the C-terminal region of PamA K278-Strep$_2$-His$_6$. The same colors in A and B corresponds to the same region in amino acid (A) and nucleotide (B) sequences.
(PDF)

**S5 Fig. Mass spectrometry analysis of PamA K278amb expressed by the pyrrolysine-based amber suppression with *p*BPa in *N. meningitidis*.** (A) Amino acid sequence of PamA with Strep$_2$-His$_6$ tag at C-terminus. The tryptic peptide containing a ncAA, *p*BPa is highlighted in red and the *p*BPa at position 278 is represented as X. (B) The incorporation of *p*BPa at position 278 was confirmed by MALDI-TOF MS analysis of the tryptic peptide IEVGXLAFSTK (X represents *p*BPa). The observed (obsd) molecular masses agreed well with the calculated (calcd) masses. (C) MALDI-TOF MS/MS analysis of the tryptic peptide of shown in B. Tandem mass spectrum of the peptide IEVGXLAFSTK (X = *p*BPa). The sequence can be read from the annotated b (red) or y (blue) ion series.
(PDF)

**S6 Fig. Examination of purified MBP-ΔN-PamA K278(*p*BPa) and Gln$_6$-ΔN-PilE recombinant proteins from *E. coli* strains for LC-MS/MS analysis.** Recombinant proteins crosslinked in *E. coli* were purified with amylose resin and concentrated to 500 μl as described in the Materials and Methods. Aliquots (approximately 1 μl) were fractionated by SDS-PAGE and analyzed by staining with a CBB staining kit (A), and Western blotting with anti-MBP (B) and anti-His$_6$ (C) mAbs. Prestained molecular mass standards are shown by grey numbers with asterisks, since the apparent molecular masses were different from the actual masses due to abnormal mobility in the gel. +/- indicates treatment or non-treatment with UV irradiation, respectively. Black, blue and red arrows indicate PamA ΔN-K278amb (no pyrrolysine-based amber suppression with *p*BPa), PamA ΔN-K278(*p*BPa)-Strep$_2$-His$_6$ (full-length) and the complex crosslinked between PamA K278(*p*BPa)-Strep$_2$-His$_6$ and ΔN-Gln$_6$-PilE, respectively. (D) Schematic figure of the crosslinking between PamA ΔN-K278(*p*BPa)-Strep$_2$-His$_6$ and Gln$_6$-ΔN-PilE recombinant proteins in *E. coli*.
(PDF)

**S7 Fig. Analysis of amino acid residues (region) in PilE crosslinked to PamA K278 (*p*BPa) by LC-MS/MS.** (A) Liquid chromatogram of uncrosslinked MBP-ΔN-PamA K278(*p*BPa) and the Gln$_6$-ΔN-PilE protein (upper), and the complex crosslinked between MBP-ΔN-PamA K278(*p*BPa) and Gln$_6$-ΔN-PilE (lower). The fraction eluted at 12.38 min from the uncrosslinked sample contained the peptide IEVGK(*p*BPa)LAFSTK corresponding to position 639 to 650 of MBP-ΔN-PamA K278(*p*BPa) (positions 274 to 284 for native PamA), determined by an MS analysis. (B) Enlarged liquid chromatogram after 14 min of elution. The fraction eluted at

17.32 min from the crosslinked sample contained the peptide MQQQQQQFTLIELMIVIA**I**V GILAAVALPAYQDYTARAQVSEAILLAEGQK, corresponding to the first 51 amino acids of Gln$_6$-ΔN-PilE at the N terminus, which was crosslinked to the peptide IEVGK($p$BPa)LAFSTK from the MBP-ΔN-PamA K278($p$BPa) protein. (C) MS spectrum of the LC fraction eluted at 17.32 min that contained the cross-linked peptide between MQQQQQQFTLIELMIVIA**I**VGI LAAVALPAYQDYTARAQVSEAILLAEGQK from Gln$_6$-ΔN-PilE and IEVGK($p$BPa)LAFSTK from MBP-ΔN-PamA K278($p$BPa)(obsd:1380.9367 [M+H]$^{5+}$).
(PDF)

**S8 Fig. MS/MS spectrum of the peptide IEVGK($p$BPa)LAFSTK from MBP-ΔN-PamA K278($p$BPa).** The sequence was read from the annotated b and y ion series; b3, y4, y5, y6, y7, y8, y9, and y10 ions were observed.
(PDF)

**S9 Fig. MS/MS spectrum of the cross-linked peptide between MQQQQQQFTLIELMIVIAI VGILAAVALPAYQDYTARAQVSEAILLAEGQK from Gln$_6$-ΔN-PilE and IEVGK($p$BPa) LAFSTK from MBP-ΔN-PamA K278($p$BPa)(obsd:1380.9367 [M+H]$^{5+}$).** The sequence was read from the annotated b and y ion series; b3, y4, y5, y6, y7, y8, y9, and y10 ions of the peptide IEVGK($p$BPa)LAFSTK from MBP-ΔN-PamA K278($p$BPa) (shown in red), and b3, b6, b9, b10, b11, b12, b13, b14, b15, b16, b17, b18, y4, y5, y6, y7, y8, y9, y10, y11, y12, y13, y14, y15, y16, y17, y18, y19, y20, y21, y22, y23, y24, y25, y26, y27, y28, y29, y30, y31, and y32 ions of the peptide MQQQQQQFTLIELMIVIA**I**VGILAAVALPAYQDYTARAQVSEAILLAEGQK from Gln$_6$-ΔN-PilE (shown in blue) were observed.
(PDF)

**S10 Fig. Sequence alignment between neisserial PilE and crystal structure of full-length gono-coccal PilE.** (A) Amino acid sequence alignment of PilE proteins from *N. meningitidis (Nm)*, *N. gonorrhoeae (Ng)* and *N. lactamica (Nl)*. The N-terminal half of the α-helix (α1-N) responsible for pilin assembly is shown by red (residues 1–14; α1:1–14) and cyan (residues 15–28; α1:15–28) bars, respectively. The αβ-loop, protruding from the globular domain to form a ridge on the sub-unit surface is indicated as a yellow bar. The D-region containing the hypervariable loop that pro-trudes as a second ridge on the globular domain is indicated as a light green bar. Red numbers indicate the positions of amino acids at which the functions of α1-N are divided. Ile at position 12 is indicated by a red vertical arrow. *N. meningitidis* HT1125 PilE (DDBJ accession no. AB698857), *N. meningitidis* H44/76 PilE (GenBank accession no. CP002420), *N. gonorrhoeae* MS11 PilE (EMBL accession no. CAI08338) and *N. lactamica* NLA_1780 type IV pilus assembly protein PilA (GenBank accession no. FN995097). (B) X-ray crystal structure of gonococcal PilE [91]. α1:1–14, α1:15–28, αβ-loop and D-region are shown in red, cyan, yellow and light green, respectively (the colors in B correspond to those in A). Ile at position 12 is indicated as a space-filling molecule.
(PDF)

**S11 Fig. Characterization of pili.** (A) Electron micrographs showing piliation of *N. meningiti-dis* strains HT1125 (wild-type), HT1822 (*ΔpamA*), HT1744 (*pilE$^+$-cat*), and HT2167 (*pilE I12A-cat*). Upper and lower panels show magnifications of 20,000 and 50,000, respectively. Scale bars shown in black (upper) and white (lower) represent 200 nm. (B) Quantification of the compe-tence for DNA transformation in *N. meningitidis* wild-type, and *pilE$^-$*, *ΔpamA*, *pilE$^+$-cat* and *pilE I12A-cat* mutants. Equivalent numbers of recipient cells were transformed using 0.5 µg of chromosomal DNA purified from the Nal$^R$ (nalidixic acid resistant) strain HT1001, and Nal$^R$ transformants were counted. Results are expressed as numbers of Nal$^R$ transformants per 1 µg DNA and ± standard deviation from at least 8 independent experiments. (C) Aggregation as assessed by phase-contrast microscopy. Aggregates of *N. meningitidis* strains HT1125 (wild-

type), HT1156 (*pilE*⁻), HT1822 (*ΔpamA*), HT1744 (*pilE*⁺*-cat*) and HT2167 (*pilE I12A-cat*) were observed after 4 hours of incubation in RPMI containing 10% fetal bovine serum.
(PDF)

**S12 Fig. Detection of homomeric crosslinking between two MBP-ΔN-PamA K278(*p*BPa).**
MS/MS spectrum of the peptide IEVGK(*p*BPa)LAFSTK (at positions 274 to 284 for native PamA) crosslinked to LNELVNLVTDLQIGAFINPNGSIAPS (at positions 247 to 273 for native PamA) in MBP-ΔN-PamA K278(*p*BPa). The crosslinked site was analyzed by SIM-XL (http://patternlabforproteomics.org/sim-xl). The sequence can be read from the annotated b and y ion series: b3, y3, y4, y5, y6 ions of the peptide IEVGK(pBPa)LAFSTK from MBP-ΔN-PamA K278(*p*BPa) (shown in red), and b3, b4, b5, b6, b7, b8, b10, b13, b19, b21, y4, y5, y6, y7, y8, y9, y10, y11, y12, y13, y14, y15 ions of a peptide LNELVNLVTDLQIGAFINPNG SIAPS from MBP-ΔN-PamA K278(*p*BPa) (blue) were observed, respectively. We could not confirm whether *p*BPa at position 278 was crosslinked to Leu at position 257 or Gln at position 258 (shown in yellow overlay).
(PDF)

**S13 Fig. Phylogenetic tree showing the relation of *pamA* genes among neisserial species: *N. meningitidis, N. gonorrhoeae* and *N. lactamica*.** The *pamA* genes (the genes encoding "conserved hypothetical protein") from three neisserial species were analyzed by Molecular Evolutionary Genetics Analysis (MEGA) X [92]. The evolutionary history was inferred using the Neighbor-Joining method [93]. The tree is drawn to scale, with branch lengths in the same units as those of the evolutionary distances used to infer the phylogenetic tree. The evolutionary distances were computed using the Maximum Composite Likelihood method [94] and are in the units of the number of base substitutions per site. The robustness of the NJ method was tested by bootstrapping with 500 replicates of data, and the percentages are shown at the nodes. This analysis involved 19 nucleotide sequences of genes encoding *N. meningitidis* HT1125 PamA (DDBJ accession no. LC511747), *N. meningitidis* H44/76 conserved hypothetical protein (GenBank accession no. CP002420), *N. meningitidis* MC58 hypothetical protein (GenBank accession no. AE002098), *N. meningitidis* M04-240196 conserved hypothetical protein (GenBank accession no. CP002423), *N. meningitidis* alpha14 conserved hypothetical protein (GenBank accession no. AM889136), *N. meningitidis* NZ-05/33 conserved hypothetical protein (GenBank accession no. CP002424), *N. meningitidis* M01-240149 conserved hypothetical protein (GenBank accession no. CP002421), *N. meningitidis* alpha710 hypothetical protein (GenBank accession no. CP001561), *N. meningitidis* 053442 conserved hypothetical protein (GenBank accession no. CP000381), *N. meningitidis* 8013 conserved hypothetical protein (GenBank accession no. FM999788), *N. meningitidis* M01-240355 conserved hypothetical protein (GenBank accession no. CP002422), *N. meningitidis* 510612 hypothetical protein (GenBank accession no. CP007524), *N. meningitidis* WUE2694 conserved hypothetical protein (GenBank accession no. FR774048), *N. meningitidis* G2136 conserved hypothetical protein (GenBank accession no. CP002419), *N. meningitidis* Z2491 hypothetical protein (GenBank accession no. AL157959), *N. meningitidis* FAM18 hypothetical protein (GenBank accession no. AM421808), *N. gonorrhoeae* FA1090 hypothetical protein (RefSeq accession no. NC_002946), *N. gonorrhoeae* NCCP11945 conserved hypothetical protein (GenBank accession no. CP001051) and *N. lactamica* putative secreted protein (GenBank accession no. FN995097).
(PDF)

**S1 Table. Oligonucleotides used in this study.**
(DOCX)

**S2 Table. Plasmids to construct *N. meningitidis* mutants.**
(DOCX)

**S3 Table. Plasmids for production of protein recombinants expressed in *E. coli*.**
(DOCX)

**S4 Table. IncQ plasmids for *N. meningitidis* strains which results were shown in supporting information.**
(DOCX)

**S5 Table. Analysis of Pili components by Tandem Mass Tag labeling.** Pili isolated from *N. meningitidis* strains HT1125 (wild-type) and HT1822 (Δ*pamA*) were subjected to trypsin digestion. Labelling of the peptides and analysis by LC-MS/MS were performed as described previously [38]. Proteins were identified by running MASCOT against the NCBI database (NCBInr 20160202). Ratio of the protein levels were expressed as the value of relative amount in HT1822 divided into those in HT1125. The results from four independent experiments were shown in this table.
(DOCX)

**S1 File.**
(PDF)

## Acknowledgments

We thank Dr. Masatomo Morita for advice about phylogenetic analyses.

## Author Contributions

**Conceptualization:** Hideyuki Takahashi, Tatsuo Yanagisawa.

**Data curation:** Hideyuki Takahashi, Tatsuo Yanagisawa.

**Formal analysis:** Hideyuki Takahashi, Naoshi Dohmae, Tatsuo Yanagisawa.

**Funding acquisition:** Hideyuki Takahashi, Shigeyuki Yokoyama.

**Investigation:** Hideyuki Takahashi.

**Methodology:** Hideyuki Takahashi.

**Project administration:** Hideyuki Takahashi, Tatsuo Yanagisawa.

**Resources:** Hideyuki Takahashi, Kwang Sik Kim, Shigeyuki Yokoyama, Tatsuo Yanagisawa.

**Software:** Hideyuki Takahashi.

**Supervision:** Hideyuki Takahashi, Tatsuo Yanagisawa.

**Validation:** Hideyuki Takahashi.

**Visualization:** Hideyuki Takahashi.

**Writing – original draft:** Hideyuki Takahashi.

**Writing – review & editing:** Hideyuki Takahashi, Kwang Sik Kim, Ken Shimuta, Makoto Ohnishi, Tatsuo Yanagisawa.

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
