## [Decision Letter · Decision Letter 0]

18 May 2020

PONE-D-20-12310

Successful genetic incorporation of non-canonical amino acid photocrosslinkers in Neisseria meningitidis: The new method shed light into the physiological function of a function-unknown NMB1345 protein in Neisseria meningitidis.

PLOS ONE

Dear Dr. Takahashi,

Thank you for submitting your manuscript to PLOS ONE. After careful consideration, we feel that it has merit but does not fully meet PLOS ONE’s publication criteria as it currently stands. Therefore, we invite you to submit a revised version of the manuscript that addresses the points raised during the review process.

After studying your manuscript, this academic editor found it confusing, poorly presented and not properly structured with lot of typos, style and grammatical errors. The both reviewers (independently) shared my view that the manuscript is in part difficult to comprehend and whole subject still requires to be substantially worked out (matured). Although one reviewer can even envisage a publication of the manuscript with the "minor revision"– the second reviewer opted for the "major revision" and expects a significant improvement in both data presentation and rewriting of the manuscript. After balancing their comments and the literature-check, it is difficult to escape the fact that the work still needs substantial further improvements. Thus, we feel that it has merit, but is not suitable for publication as it currently stands. Therefore, my decision is "Major Revision." We hope that criticism of the experts will help to improve the quality of your revised manuscript.

We would appreciate receiving your revised manuscript by Jul 02 2020 11:59PM. To enhance the reproducibility of your results, we recommend that if applicable you deposit your laboratory protocols in protocols.io, where a protocol can be assigned its own identifier (DOI) such that it can be cited independently in the future. For instructions see: http://journals.plos.org/plosone/s/submission-guidelines#loc-laboratory-protocols

We look forward to receiving your revised manuscript.

Kind regards,

Ned Budisa

Academic Editor

PLOS ONE

Journal Requirements:

Additional Editor Comments:

I asked highly experted reviewers and observers in the field for a fair, unbiased and balanced review on your paper. Now their comments are available to you. While the referee 2 justifiably voiced general concerns about the (a) scientific style (manuscript structure and focus), (b) practical relevance and (c) good scientific presentation in general, the referee 1 had many highly relevant technical issues which urgently needed to be addressed. All these concerns must be addressed correctly (point-to-point) in the revised manuscript (only when these conditions are fully met, I can handle the manuscript again as an academic editor). Obviously, the work still needs substantial improvements. I feel that it has merit, but is not suitable for publication as it currently stands. Therefore, my decision is again "Major Revision" and I invite you to submit a revised version of the manuscript that addresses the points raised as elaborated above.

Reviewers' comments:

Reviewer's Responses to Questions

**Comments to the Author**

1. Is the manuscript technically sound, and do the data support the conclusions?

Reviewer #1: Yes

Reviewer #2: Partly

2. Has the statistical analysis been performed appropriately and rigorously? 

Reviewer #1: Yes

Reviewer #2: I Don't Know

3. Have the authors made all data underlying the findings in their manuscript fully available?

Reviewer #1: Yes

Reviewer #2: Yes

4. Is the manuscript presented in an intelligible fashion and written in standard English?

Reviewer #1: Yes

Reviewer #2: No

5. Review Comments to the Author

Reviewer #1: The manuscript has applied the genetic code expansion tools to reveal interaction of a hypothetical protein, NMB1345, to the PilE, a major component of meningococcal pili, in Neisseria meningiditis. Introduction of a new methodology to identify possible function of unknown factors in pathogenic bacteria is an interesting subject for further application of the orthogonal translation systems and create a new area of research. The manuscript has been prepared and written in scientific manner and just some minor issues has to be remained which mentioned below:

1- The quality of figures are very low and should be improved.

2- The number of figures is too high. Some of them could be moved in “Supporting information” part. For example, Fig. 4 and Fig. 7. Moreover, figures 5 and 6 should be combined.

3- There is some mistyping in the text. For example: line 86, “intarected!!”; line 110, “intection!!”; line 149 in Table 1, readjusting needed; line 635, double typing or moving after ref.; line 737, “seemd!!”

4- Line 122: “L plates”? Clarify.

5- Line 803: “proteins”; only one protein highlighted in red, but in the text two proteins mentioned. Delete “s” or highlight the second one (NMEN93004-1215).

6- In Fig. 3B: Western blotting would not be able to confirm proper incorporation of ncAA. The MS analysis should be done. Please clarify this issue in corresponding part.

7- The last but not least, please explain how chemically a protein like MBP-�N-PamA K278(pBPa) crosslinked to the endogenous proteins like Gln6-�N-PilE? Self-crosslinking speculation is possible but crosslinking with the accompanying proteins in vivo needs to be clarified in the text. A schematic illustration would be helpful.

Reviewer #2: Abstract:

“…Advances have been achieved in many technologies, and one of the most developed methods is genetic code expansion with non-canonical amino acids (ncAAs) utilizing a pyrrolysyl-tRNA synthetase/ tRNA Pyl pair from Methanosarcina species;…”

What does most developed methods mean? This could just mean something with a reference system, otherwise it contains no information.

“… In the present study, we developed the new method to genetically incorporate ncAAs into N. meningitides…”

How can one introduce ncAAs into an organism? Maybe into a protein sequence?

“…In the present study, we developed the new method to genetically incorporate ncAAs into N. meningitidis and elucidate the biological function(s) of the NMB1345 protein in N. meningitidis by using ncAA-encoded photocrosslinking probes…”

Why extend the sentence unnecessarily? Say that your ncAAs are photcrosslinker. For example: In the present study, we developed the new method to genetically incorporate ncAA photocrosslinking probes into N. meningitidis and elucidate the biological function(s) of the NMB1345 protein in N. meningitidis

Introduction:

Line 42-45: “However, N. meningitidis exhibits the ability to cross the epithelial layer of the upper

43 respiratory tract, infiltrate the bloodstream, evade the defenses of the human immune

44 system, adhere to the endothelial layers of peripheral and brain vessels, cross the brain45

blood barrier, and replicate in cerebrospinal fluid.”

Citation is missing.

Line 84-86: “We previously attempted using already existing methods, such as the two-hybrid system or a pull-down assay, to identify the proteins intarected with NMB1345 protein, but were unsuccessful.”

Interacted is written wrong, also what does proteins interacted with NMB 1345 mean? Maybe protein-protein interaction?

Line 93: E. coli and C. elegans need to be written in full name. No full name before. Also C. elegans is written wrong.

Line 88-94: Needs rephrasing. The sentence is way to long. In addition, genetic code expansion with the Pyl system is anything but recently developed.

Line 97 99: No citation for the last claim in the sentence

Line 103-104: “... most useful …” this statement makes just sense if you say in what regard it is most useful. Nothing can be useful without stating the purpose.

Line 110: intection?

Line 111-112: If you state what PamA means, then do it the first time you use it, that would be in line 110.

Line 122: What are L plates or L broth? Please define or state the company.

Line 551: This sentence structure makes no sense. Your use of “and” implies that you genetically characterized the gene product. What does this mean? You did fluorescence.

Line 554: This is an interpretation of the data, normally in the text. In the figure caption one should describe what method was used and what was measured.

Line 563: using beneath makes no sense. Maybe you meant amongst or between strains?

Line 573-577: Sentence structure makes it almost incomprehensible.

Line 610-611: Your description is the opposite what the figure is showing, which is it?

Line 614: You say you used 123 amino acids but in figure 2B it is written that you used 113. Which is it? Figure 2B and the caption for that is confusing. Please clear up which parts you used and describe it with the correct length. Adapt the main text accordingly.

Line 616-618: This part is written extremely confusing. Where does strain 2215 and 2217 come from? Was not mentioned before or clearly described. Also Figure 2C does not annotate any of theses strains. Please adapt this.

Line 619-620: Please elaborate your conclusion.

Line 620-625: These claims are not evaluable since the Figure D and E are not consistent described. Please correct your Figure as mention above.

Line 630-632: “…while no one examined whether the orthogonality of the tRNAPyl-PylRS pair is specific or compatible with the meningococcal translational system.”

This should be put into the introduction, not results.

Line 642: The general description for Fig. 3does not match what is shown. You don’t show just the successful introduction of ncAAs into proteins.

Figure 3E: The gray arrow an text is not easy visible

Line 680: I guess you mean steric and not stereological. But since you have no further information why the suppression does not work I would omit this speculation. It could be also context effects, mRNA secondary structure effects, etc.

Line 684-685: “Moreover, when N. meningitidis was irradiated with UV light, a more dimer form of GST was detected with longer irradiation,…” Please rewrite, makes no sense

Line 705-707: What does this mean? Why mention HdeA from E.coli? Please elaborate the analogy or omit the sentence.

Line 709: “There were 40 Lys residues in the PamA protein...” wrong tense. Doesn’t PamA still has 40 Lys?

Line 720: You meant whole cell irradiation? Intact strains sounds weird.

Line 737: seemed not seemd

742-745: This sentence makes no sense, rewrite.

Line 747: The general description is not correct. Also boldening is not correct. Also no time of irradiation is mentioned.

Line 750: Portion?

Line 751: What was suppressed? How can an amino acid suppress something? Used several times. Please be more precise in the use of your language.

Line 759: If the contain other proteins then these are protein complexes.

Line 765-771: Super long sentence. Rephrase for clarity.

Line 789:-791 Why is there two times referred to A and one time to B. Just one time A.

Line 805-808: Phrasing makes no sense. Rephrase.

General remarks:

Check all abbreviations of consistency. Sometime you use IM for inner membrane; sometimes you use it for inner membrane-enriched fraction.

6. PLOS authors have the option to publish the peer review history of their article (what does this mean?). If published, this will include your full peer review and any attached files.

Reviewer #1: Yes: Hamid Reza Karbalaei-Heidari

Reviewer #2: No

---

## [Author Response · Author response to Decision Letter 0]

26 Jun 2020

Jun 26, 2020

Dr. Nediijko Budisa,

PLOS ONE

Academic editor

Dear Dr. Budisa,

Thank you very much for your favorable review of our manuscript.

We have revised our manuscript according to the comments of the academic editor and two reviewers. In addition, we carefully rewrote the entire manuscript to improve the English and asked a native speaker to correct the English.

Journal Requirements:

We revised the first page of manuscript to present on title, authors, affiliations, and corresponding author according to the PLOS ONE style.

 We prepared the original image files according to these instructions. The original blots and gel images-files are as a PDF file in Supporting information, named as “Original blots and gel images”.

 We removed all of the descriptions mentioned as “data not shown”.

Review Comments to the Author

Reviewer #1: The manuscript has applied the genetic code expansion tools to reveal interaction of a hypothetical protein, NMB1345, to the PilE, a major component of meningococcal pili, in Neisseria meningiditis. Introduction of a new methodology to identify possible function of unknown factors in pathogenic bacteria is an interesting subject for further application of the orthogonal translation systems and create a new area of research. The manuscript has been prepared and written in scientific manner and just some minor issues has to be remained which mentioned below:

1- The quality of figures are very low and should be improved.

We revised all of the figures according to the PLOS ONE style, and hope the present quality of the figures is acceptable.

2- The number of figures is too high. Some of them could be moved in “Supporting information” part. For example, Fig. 4 and Fig. 7. Moreover, figures 5 and 6 should be combined.

Since we consider Fig 4B to be essential to comprehend the content of this study, we retained Fig 4B as Fig 4 and moved the previous Fig 4A to the supporting information as the new S3 Fig.

According to reviewer 1’s suggestion, the previous Figures 5 and 6 were combined into one figure as the new Fig 5, and Fig 7 was moved to the supporting information as the new S7-9 Figs.

3- There is some mistyping in the text. For example: line 86, “intarected!!”; line 110, “intection!!”; line 149 in Table 1, readjusting needed; line 635, double typing or moving after ref.; line 737, “seemd!!”

The typographic errors in lines 86,110 and 737 were corrected.

Table 1 was arranged as follows: since Fig 4A was moved to the supporting information as S3 Fig, the description of the meningococcal plasmids in the supporting information (new S3 Fig) was moved to the new S4 Table.

Double typing at line 635 was corrected.

4- Line 122: “L plates”? Clarify.

We corrected the description “L broth” and “L plate” to Luria-Bertani (LB) broth and LB plate (LB liquid medium containing 1.5% agarose). 

5- Line 803: “proteins”; only one protein highlighted in red, but in the text two proteins mentioned. Delete “s” or highlight the second one (NMEN93004-1215).

We deleted “s”.

6- In Fig. 3B: Western blotting would not be able to confirm proper incorporation of ncAA. The MS analysis should be done. Please clarify this issue in corresponding part.

As reviewer 1 commented, from the physicochemical viewpoint, as MS analysis could confirm the proper incorporation of ncAA in OpaD K95amb, GST E51amb and GST F52amb. 

On the other hand, purification of the OpaD and GST proteins from N. meningitidis strains was truly impossible since these proteins lacked tags and were expressed at comparable levels to endogenously produced proteins that could only be detected by Western blotting, as in higher organisms such as mammalian cells, Caenorhabditis elegans (ACS Chem Biol 20:1292-1302, 2012), neural cells (Stem cells 29:1231-1240, 2011) and zebrafish (Cell Research 27:294-297, 2017).

Considering of the technical difficulties, the purification of OpaD K95(mAzZLys) and OpaD K95(pBPa) from the N. meningitidis strain was impossible, and the results using OpaD K95amb shown in the former Fig 3BC were deleted in the revised manuscript.

As for GST E51(mAzZLys), GST E51(pBPa) and GST F52(pBPa), we constructed the new plasmids harboring gst genes fused to Strep2His6 tag for N. meningitidis. However, some materials required to introduce the plasmids into N. meningitidis were out of stock and we would have to wait for a month for their delivery. Moreover, even if we were able to obtain the N. meningitidis transformants, the purification of GST proteins followed by MS analysis will take approximately two months, since this is not an overproduction system. From these reasons, we could not purify GST E51(mAzZLys), GST E51(pBPa) and GST F52(pBPa) from N. meningitidis during this short period and could not prove the incorporation by MS analysis in the revised manuscript.

However, we purified the PamA K278(pBPa) protein from an N. meningitidis strain grown on approximately 200 plates (Fig 5A lane 6). We subjected the PamA K278(pBPa) protein to MS analysis and confirmed the selective incorporation of pBPa in N. meningitidis in the revised manuscript (S5 Fig).

We corrected the words “prove” or “confirm” to show that the proper incorporation of ncAA in GST E51amb and GST P52amb also occurred in N. meningitidis. 

We hope our revised results sufficiently answer to the reviewer 1’s comments.

7- The last but not least, please explain how chemically a protein like MBP-�N-PamA K278(pBPa) crosslinked to the endogenous proteins like Gln6-�N-PilE? Self-crosslinking speculation is possible but crosslinking with the accompanying proteins in vivo needs to be clarified in the text. A schematic illustration would be helpful.

In our experiments, both the MBP-�N-PamA K278(pBPa) and Gln6-�N-PilE proteins were overproduced and crosslinked in E. coli. Our poor explanation probably led to reviewer 1’s confusion. 

We revised the description to clearly show that the crosslinking between MBP-�N-PamA K278(pBPa) and Gln6-�N-PilE proteins was performed in E. coli. Moreover, a schematic illustration of these crosslinked proteins was added in S6D Fig.

We hope our revisions are suitable to answer for reviewer 1’s comments.

Reviewer #2: 

Abstract:

“…Advances have been achieved in many technologies, and one of the most developed methods is genetic code expansion with non-canonical amino acids (ncAAs) utilizing a pyrrolysyl-tRNA synthetase/ tRNA Pyl pair from Methanosarcina species;…”

What does most developed methods mean? This could just mean something with a reference system, otherwise it contains no information.

We revised the sentence by adding the following phrase “Among many biological technologies to examine transient protein-protein interactions in vivo” (Lines 24-25).

“… In the present study, we developed the new method to genetically incorporate ncAAs into N. meningitides…”

How can one introduce ncAAs into an organism? Maybe into a protein sequence?

We revised the sentence by adding the following phrase: ”encoded photocrosslinking probes into N. meningitidis by utilizing a pyrrolysyl-tRNA synthetase/tRNAPyl pair” (Lines 29-31).

“…In the present study, we developed the new method to genetically incorporate ncAAs into N. meningitidis and elucidate the biological function(s) of the NMB1345 protein in N. meningitidis by using ncAA-encoded photocrosslinking probes…”

Why extend the sentence unnecessarily? Say that your ncAAs are photcrosslinker. For example: In the present study, we developed the new method to genetically incorporate ncAA photocrosslinking probes into N. meningitidis and elucidate the biological function(s) of the NMB1345 protein in N. meningitidis

We revised the sentence according to reviewer’s 2 suggestion as ”we developed a new method to genetically incorporate ncAAs-encoded photocrosslinking probes into N. meningitidis by utilizing a pyrrolysyl-tRNA synthetase/tRNAPyl pair and elucidate the biological function(s) of the NMB1345 protein” (Lines 29-31).

Introduction:

Line 42-45: “However, N. meningitidis exhibits the ability to cross the epithelial layer of the upper

43 respiratory tract, infiltrate the bloodstream, evade the defenses of the human immune

44 system, adhere to the endothelial layers of peripheral and brain vessels, cross the brain45

blood barrier, and replicate in cerebrospinal fluid.”

Citation is missing.

This citation was added in the revised manuscript (Line 44).

Line 84-86: “We previously attempted using already existing methods, such as the two-hybrid system or a pull-down assay, to identify the proteins intarected with NMB1345 protein, but were unsuccessful.”

Interacted is written wrong, also what does proteins interacted with NMB 1345 mean? Maybe protein-protein interaction?

We revised the following phrase: “We previously attempted to examine protein-protein interaction using existing methods, such as the two-hybrid system and a pull-down assay, to find some cues on the function the NMB1345 protein” (Lines 82-85).

Line 93: E. coli and C. elegans need to be written in full name. No full name before. Also C. elegans is written wrong.

E. coli and C. elegans were rewritten as Escherichia coli and Caenorhabditis elegans, respectively (Lines 92-93).

Line 88-94: Needs rephrasing. The sentence is way to long. In addition, genetic code expansion with the Pyl system is anything but recently developed.

The sentence was divided into two parts. In addition, we removed the phrase “most recently” in the revised manuscript (Lines 87-94).

Line 97 99: No citation for the last claim in the sentence

The citation was added in the revised manuscript (Line 98).

Line 103-104: “... most useful …” this statement makes just sense if you say in what regard it is most useful. Nothing can be useful without stating the purpose.

We changed the phrase “most useful” to “most widely used” in the revised manuscript (Line 105).

Line 110: intection?

We corrected the mistyping. 

Line 111-112: If you state what PamA means, then do it the first time you use it, that would be in line 110.

We mentioned the designation of NMB1345 as PamA in lines 109-110 in the revised manuscript.

Line 122: What are L plates or L broth? Please define or state the company.

Since both reviewers pointed out the same problems, we corrected the description “L broth” and “L plate” as Luria-Bertani (LB) broth and LB plate (LB liquid medium containing 1.5% agarose) (Lines 122-123).

Line 551: This sentence structure makes no sense. Your use of “and” implies that you genetically characterized the gene product. What does this mean? You did fluorescence.

We changed the title of Fig 1 to “Characterization of �pamA N. meningitidis” (Line 540).

Line 554: This is an interpretation of the data, normally in the text. In the figure caption one should describe what method was used and what was measured.

We changed the title of Fig 1B to “Effect of �pamA mutation on N. meningitidis infection of HBMEC” (Line 543).

Line 563: using beneath makes no sense. Maybe you meant amongst or between strains?

It is well known that bacterial infection elicits alternations of the signaling pathways concomitant with cytoskeletal reorganization in the host cells, and that ezrin is one of the key molecules controlling both cell shape and signaling. N. meningitidis reportedly induce ERM accumulation beneath the bacterial colony, and this accumulation is required for bacterial internalization into host cells (Takahashi et al, . IAI 2011, 2012 and 2015). This explanation was described in lines 560 to 562. In addition, since ezrin is the host cell’s factor, we considered that “beneath” would be an appropriate word to show the part about the condensation of ezrin under the meningococcal colonies, and this word was often used in our previous manuscripts (Takahashi et al, IAI 2011, 2012 and 2015).

We hope the term “beneath” is acceptable for reviewer 2. 

Line 573-577: Sentence structure makes it almost incomprehensible.

We revised the sentence for clarification (Lines 562-564).

Line 610-611: Your description is the opposite what the figure is showing, which is it?

As reviewer 2 pointed out, the description was opposite and corrected in the revised manuscript.

Line 614: You say you used 123 amino acids but in figure 2B it is written that you used 113. Which is it? Figure 2B and the caption for that is confusing. Please clear up which parts you used and describe it with the correct length. Adapt the main text accordingly.

We mistyped 23 as “123” in the previous version. PamA consists of the first 3 amino acids (aa) residues, the putative membrane-spanning region (20aa) and the hydrophilic region (493aa) as shown in Fig 2B. On the other hand, in the experiments with lacZ or phoA fusion, the first 136 aa were fused to the lacZ or phoA gene to genetically examine the orientation of PamA in N. meningitidis. We corrected the number.

Line 616-618: This part is written extremely confusing. Where does strain 2215 and 2217 come from? Was not mentioned before or clearly described. Also Figure 2C does not annotate any of these strains. Please adapt this.

The explanations of HT2215 and HT2217 are described in the text. To prevent confusion, the two strains are referred to in Table 2 and the strain names are also shown in Fig 2B.

Line 619-620: Please elaborate your conclusion.

Genetical methods are widely used to determine the orientation of a protein relative to the inner membrane in bacteria by expressing LacZ and PhoA fusion proteins, since the alkaline phosphatase activity of PhoA is active only in the periplasmic space (Boyle-Vavra et al, 1995). The explanation is added in Lines 608-609.

Line 620-625: These claims are not evaluable since the Figure D and E are not consistent described. Please correct your Figure as mention above.

As answered above, the descriptions of Fig 2 D and E were opposite in the text and corrected in the revised manuscript.

Line 630-632: “…while no one examined whether the orthogonality of the tRNAPyl-PylRS pair is specific or compatible with the meningococcal translational system.”

This should be put into the introduction, not results.

The sentence “while no one examined whether the orthogonality of the tRNAPyl-PylRS pair is specific or compatible with the meningococcal translational system” was moved into the Introduction (Lines 98-100) in the revised manuscript.

Line 642: The general description for Fig. 3does not match what is shown. You don’t show just the successful introduction of ncAAs into proteins.

Figure 3E: The gray arrow an text is not easy visible

As reviewer 2 pointed out, the title of Fig 3 was not suitable for the content. We corrected the tile as “Incorporation of ncAAs into N. meningitidis proteins monitored by Western blotting” (Line 630). 

Line 680: I guess you mean steric and not stereological. But since you have no further information why the suppression does not work I would omit this speculation. It could be also context effects, mRNA secondary structure effects, etc.

As reviewer 2 pointed out, there are some reasons for the failure of mAzZLys incorporation at GST F52amb. However, we do not know the reason for the failure. To eliminate unnecessary confusion, the description of the steric hinderance was removed in the revised manuscript (Line 646).

Line 684-685: “Moreover, when N. meningitidis was irradiated with UV light, a more dimer form of GST was detected with longer irradiation,…” Please rewrite, makes no sense

We rewrote the sentence as “longer UV irradiation increased the amount of the dimer form of GST” (Line 658).

Line 705-707: What does this mean? Why mention HdeA from E.coli? Please elaborate the analogy or omit the sentence.

As reviewer 2 suggested the description of the E. coli HdeA protein was deleted in the revised manuscript.

Line 709: “There were 40 Lys residues in the PamA protein...” wrong tense. Doesn’t PamA still has 40 Lys?

As shown in S1 Fig, there are 40 Lys residues (shown in red) in the deduced amino acid sequence of the PamA protein.

Line 720: You meant whole cell irradiation? Intact strains sounds weird.

We rewrote the phrase “intact” as “whole bacterial cells” (Line 690).

Line 737: seemed not seemd

We corrected the mistyping.

742-745: This sentence makes no sense, rewrite.

We rewrote the sentence as “In the present study, we focused only on the proteins crosslinked to PamA K278(pBPa) because the three bands with the highest intensities were found in the PamA K278(pBPa) sample”.

Line 747: The general description is not correct. Also boldening is not correct. Also no time of irradiation is mentioned.

Since we revised Fig 4 according to the reviewer 1’s suggestion, the title for the new Fig 4 was rewritten in the revised manuscript.

This is the title of Fig 4, which is now shown in bold according to the guidelines. 

PamA K208(pBPa), PamA K278(pBPa), PamA K314(pBPa), PamA K382(pBPa) and PamA K395(pBPa) proteins crosslinked to unidentified endogenous proteins in N. meningitidis. (Line 715-717)

While the irradiation time is described in the Materials and Methods, the irradiation time is also described in the figure citation (Line 720).

Line 750: Portion?

 We rewrote the word as “aliquot” in the revised manuscript (Line 721)

Line 751: What was suppressed? How can an amino acid suppress something? Used several times. Please be more precise in the use of your language.

We meant to use the phrase “suppressed” in the genetic meaning. However, to exclude confusion, all phrases with “suppressed” in the old manuscript were rewritten as “expressed by pyrrolysine-based amber suppression with pBPa” in the revised manuscript.

Line 759: If the contain other proteins then these are protein complexes.

According to reviewer’s 2 suggestion, we rewrote the phrase “proteins” as “protein complexes” (Line 726).

Line 765-771: Super long sentence. Rephrase for clarity.

Since the explanation of Twin-strep-tag was cited in the text (Korndorfer and Skerra, 2002), we shortened the sentence by deleting the explanation (Lines 727-728).

Line 789:-791 Why is there two times referred to A and one time to B. Just one time A.

According to reviewer 1’s suggestion, we combined the previous Figs 4 and 5 into the new Fig 5. The revision changed the figure citations of the new Fig 5.

We corrected the reference to (A) in the figure citation of the new Fig 5.

Line 805-808: Phrasing makes no sense. Rephrase.

We rewrote the long sentences as follows: We then characterized the 85-kDa band, which could contain approximately 20 kDa endogenous protein crosslinked to PamA K278(pBPa). LC-MS/MS followed by MASCOT analyses revealed that two possible candidates, a fimbrial protein (17 kDa) and the hypothetical protein NMEN93004_1215 (18 kDa) (Table 3)(Lines 786-789).

General remarks:

Check all abbreviations of consistency. Sometime you use IM for inner membrane; sometimes you use it for inner membrane-enriched fraction.

We checked all of the abbreviations used in this study. As for the “IM”, we intended IM to mean Inner membrane, but not the Inner-membrane-rich fraction, and corrected it for clarity.

 All figures were converted by PACE.

We appreciate your kind consideration of this revised manuscript and are looking forward to hearing a favorable response from you.

Sincerely yours,

Hideyuki Takahashi, Ph.D.

 Department of Bacteriology I

 National Institute of Infectious Diseases

 Toyama 1-23-1, Shinjuku

 Tokyo 162-8640, Japan

---

## [Decision Letter · Decision Letter 1]

5 Aug 2020

Genetic incorporation of non-canonical amino acid photocrosslinkers in Neisseria meningitidis: New method provides insights into the physiological function of the function-unknown NMB1345 protein.

PONE-D-20-12310R1

Dear Dr. Takahashi,

The paper is critically evaluated by two highly experted reviewers in the field that are capable of a fair, unbiased and balanced review. As you can see from their comments, both reviewers agreed that the data in the manuscript are now better presented and manuscript is acceptable for publication with minor improvements.We’re pleased to inform you that your manuscript has been judged scientifically suitable for publication and will be formally accepted for publication once it meets all outstanding technical requirements.

Kind regards,

Nediljko Budisa

Academic Editor

PLOS ONE

Additional Editor Comments (optional):

Reviewers' comments:

Reviewer's Responses to Questions

**Comments to the Author**

1. If the authors have adequately addressed your comments raised in a previous round of review and you feel that this manuscript is now acceptable for publication, you may indicate that here to bypass the “Comments to the Author” section, enter your conflict of interest statement in the “Confidential to Editor” section, and submit your "Accept" recommendation.

Reviewer #1: All comments have been addressed

Reviewer #2: All comments have been addressed

2. Is the manuscript technically sound, and do the data support the conclusions?

Reviewer #1: Yes

Reviewer #2: Yes

3. Has the statistical analysis been performed appropriately and rigorously? 

Reviewer #1: Yes

Reviewer #2: Yes

4. Have the authors made all data underlying the findings in their manuscript fully available?

Reviewer #1: Yes

Reviewer #2: Yes

5. Is the manuscript presented in an intelligible fashion and written in standard English?

Reviewer #1: Yes

Reviewer #2: Yes

6. Review Comments to the Author

Reviewer #1: (No Response)

Reviewer #2: (No Response)

7. PLOS authors have the option to publish the peer review history of their article (what does this mean?). If published, this will include your full peer review and any attached files.

Reviewer #1: **Yes: **Hamid Reza Karbalaei-Heidari

Reviewer #2: No

---

## [Editor Report · Acceptance letter]

7 Aug 2020

PONE-D-20-12310R1 

Genetic incorporation of non-canonical amino acid photocrosslinkers in Neisseria meningitidis: New method provides insights into the physiological function of the function-unknown NMB1345 protein. 

Dear Dr. Takahashi:

I'm pleased to inform you that your manuscript has been deemed suitable for publication in PLOS ONE. Congratulations! Your manuscript is now with our production department. 

Kind regards, 

on behalf of

Prof. Dr. Nediljko Budisa 

Academic Editor

PLOS ONE